# A global scale evaluation of extreme event uncertainty in the *eartH2Observe* project

Toby R. Marthews[1], Eleanor M. Blyth[1], Alberto Martínez-de la Torre[1], Ted I. E. Veldkamp[2]

[1]Centre for Ecology & Hydrology, Maclean Building, Wallingford OX10 8BB, U.K.
5   [2]Institute for Environmental Studies, Vrije Universiteit Amsterdam, 1081 HV Amsterdam, The Netherlands

*Correspondence to*: Toby R. Marthews (tobmar@ceh.ac.uk)

**Abstract.** Knowledge of how uncertainty propagates through a hydrological land surface modelling sequence is of crucial importance in the identification and characterisation of system weaknesses in the prediction of droughts and floods at global scale. We evaluated the performance of five state-of-the-art global hydrological and land surface models in the context of modelling extreme conditions (drought and flood). Uncertainty was apportioned between model used (model skill) and also the satellite-based precipitation products used to drive the simulations (forcing data variability) for extreme values of precipitation, surface runoff and evaporation. We found in general that model simulations acted to augment uncertainty rather than reduce it. In percentage terms, the increase in uncertainty was most often less than the magnitude of the input data uncertainty, but of comparable magnitude in many environments. Uncertainty in predictions of evapotranspiration lows (drought) in dry environments was especially high, indicating that these circumstances are a weak point in current modelling system approaches. We also found that high data and model uncertainty points for both ET lows and runoff lows were disproportionately concentrated in the equatorial and southern tropics. Our results are important for highlighting the relative robustness of satellite products in the context of land surface simulations of extreme events and identifying areas where improvements may be made in the consistency of simulation models.

## 1 Introduction

Producing robust predictions about the future dynamics of the water cycle at local, regional and global scales is critically important because it is the only way to avoid or mitigate the effects of water cycle extremes (e.g. flood, drought) (IPCC, 2012) and, in the longer term, to improve our use of resources and achieve long-term adaptation to climate change (Bierkens, 2015). Over the 21st century, climate and hydrological regimes are predicted to undergo significant shifts in baseline variables such as temperature, precipitation and runoff, leading to changes in the frequency of extremes of precipitation, evaporation and overland flow, and ultimately to changes in the frequency and intensity of both floods and droughts (Bierkens, 2015; Dadson et al., 2017; Marthews et al., 2019; Prudhomme et al., 2014). Understanding and predicting these shifts in the global dynamical system, both at atmospheric and land surface level is therefore of crucial importance (Santanello *et al.* 2018).

All model predictions have uncertainties, and linked modelling sequences have identifiable uncertainties at each step in the sequence (uncertainty propagation). In the case of a hydrological land surface modelling sequence, where climate data inputs are used to drive a simulator of the surface water cycle and land surface interactions, there are two main sources of uncertainty: *data uncertainty* (differences between forcing data used) and *model uncertainty* (differences between the simulation models). Data and model uncertainty differ greatly not just between themselves at particular locations, but also between coastal and floodplain areas of the world, and remote regions with heterogeneous terrain (Bhuiyan et al., 2018a; Riley et al., 2017) and between extreme high flows (floods) (Mehran and AghaKouchak, 2014; Nikolopoulos et al., 2016) and extreme water scarcity (droughts) (Veldkamp and Ward, 2015).

We focus on the relative dominance of model uncertainty (we take this as a broadly defined measure, including uncertainty from hydrology models that simulate water dynamics, vegetation models that focus on carbon dynamics and land surface models that attempt to integrate all biogeochemical cycles) and uncertainty in the precipitation product used to drive those models. In situations where model uncertainty is significant, the range of predictions possible from standard model simulations is of great importance to stakeholders and other users. If precipitation data uncertainty dominates, however, then greater attention should arguably be focused on selecting the most appropriate product to use, and perhaps additionally on interrogating the potentially sparse data base of precipitation measuring stations used by the precipitation products.

## 1.1 Uncertainties in land surface model simulations

Model uncertainty, i.e. prediction variation as a result of differing process representations within a model (e.g. Li and Wu (2006)), is commonly the dominant uncertainty in complex systems used in risk-informed decision-making (Oberkampf and Roy, 2010). Although historically often overlooked (Li and Wu, 2006), model uncertainty has recently come under increasing scrutiny in the context of land surface models (Huntingford et al., 2013; Long et al., 2014; Schewe et al., 2014; Ukkola et al., 2016). A lack of adequate representation of flood-generation processes (both from surface and subsurface runoff) and permafrost or snow dynamics can lead to an imprecise simulation of runoff peaks in many large river basins, and a lack of proper representation of wetland evaporation and human effects such as water consumption and inter-basin transfers can lead to over- or under-estimated discharge in many basins, especially those with large semiarid regions (Bierkens, 2015; Veldkamp et al., 2018). Additionally, even though regional-scale precipitation is predominantly caused by the atmospheric moisture convergence associated with large-scale and mesoscale circulations, processes operating on smaller length scales significantly modify even regional-scale dynamics, so it is to be expected that uncertainty in land surface models will depend on local topography, the presence or absence of vegetation or water bodies and, importantly, which type of precipitation is dominant at a particular point and time (cyclonic, orographic or convective, Table 1).

## 1.2 Uncertainties in precipitation products

Precipitation is a necessary forcing input for land surface and hydrological models that is extremely challenging to estimate independently (Beck et al., 2017b; Bhuiyan et al., 2018a; Bhuiyan et al., 2018b; Levizzani et al., 2018). The accuracy and

precision of precipitation measurements fundamentally influences predictions of land surface and hydrological models (Hirpa et al., 2016), however many widely-used precipitation products have high uncertainties over the tropics and/or areas of high relief (Bierkens, 2015; Derin et al., 2016; Kimani et al., 2017; Yin et al., 2015).

High precipitation extremes are not always well-characterised: Mehran and AghaKouchak (2014) reviewed the capabilities of satellite precipitation datasets to estimate heavy precipitation rates at different temporal accumulations. For example, the precipitation radar on board TRMM (Table 2) is capable of capturing moderate to heavy precipitation but does not detect light rain or drizzle (Huffman et al., 2007; Luo et al., 2017).

Low precipitation extremes are also not always well-characterised: Veldkamp and Ward (2015) reviewed the advantages of different drought indices and highlighted many issues at the global scale. This relates to a more general point about remote sensing rainfall intensity: a precipitation product is more likely to record correctly that it is raining at a particular location than to record correctly the amount, which is unfortunate because it is usually precipitation amount that is most important for predictive modelling of drought or flood intensity.

Accuracy of meteorological data including precipitation will be expected to be lower (and uncertainty higher) for 'real-time' precipitation products because they have not been 'blended' with raingauge or reanalysis data (Table 2) (Munier et al., 2018). If a near-real time estimate of drought or flood is needed, therefore, then a cost-benefit balance arises with the end user having to make a choice between up-to-date information versus lowest uncertainty (Munier et al., 2018).

## 1.3 The *eartH2Observe* project

During 2014-2018, the *eartH2Observe* project http://www.eartH2Observe.eu/ brought together a multinational team of modelling and Earth Observation (EO) researchers to improve the assessment of global water resources through the integration of new datasets and modelling techniques. The uncertainties described above for different parts of the forcing data - land surface model system have been the starting point for this investigation, and *eartH2Observe* has quantified these uncertainties using an ensemble of forcing data and modelling systems. The project aimed to provide an overall understanding of the uncertainty in the EO products and EO-driven water resources models. This understanding is needed for optimal data-model integration and for water resources reanalysis, and their use for basin scale and end-user applications (e.g. floods, droughts, basin water budgets, stream flow simulations) (Nikolopoulos et al., 2016). As part of *eartH2Observe*, and in order to make progress towards this aim, in this study we asked the following two research questions:

(1) Under what circumstances can uncertainty in the prediction of water cycle quantities be attributed clearly to the model in use (model uncertainty) and/or to the precipitation product used to drive the model (data uncertainty)?

(2) When uncertainty is attributable to both model and data sources, is data uncertainty generally the greater (i.e. the model contributes less than 50% of total uncertainty) or the lesser?

## 2 Data and methods

Uncertainty in extreme event representation varies both between models used (model uncertainty) and also between satellite-based precipitation products used to drive the simulations (data uncertainty). Five of the most widely-used and well-supported precipitation data products were used in this study (Table 2) and five state-of-the-art land surface models and hydrological models were run using each of those forcing data products (Table 3). This produced an ensemble of 25 estimates for each output variable.

Only the precipitation forcing data for each model were allowed to vary between simulations: the remaining non-precipitation drivers (temperature, wind speed, radiation, etc.) were held constant across all simulations and taken from global Water Resources Reanalysis 2 baseline forcing data used in other *eartH2Observe* projects (WRR2) (Arduini et al., 2017). The combination of WRR2 non-precipitation drivers and the selected precipitation drivers (Table 2) is called WRR-ENSEMBLE (Arduini et al., 2017). All simulations used a global spatial resolution of 0.25° and covered the period 2000-2013. Because of source data limitations (Table 2), we restricted our analysis to latitudinal zones between 50°S and 50°N (Fig. 1).

### 2.1 Focus on extremes

Performance was assessed in terms of the variability of evapotranspiration (ET) and surface runoff under extreme rainfall conditions (both high extremes and low extremes). We quantified the relative magnitudes of these uncertainties under (i) varying simulation model (model uncertainty) and (ii) varying choice of precipitation product (data uncertainty). We quantified uncertainty in terms of the number of extreme events per month, with the *extreme event* defined as the occurrence of an extreme value for the monthly average of a given variable, and *extreme* defined as a value in the top/bottom 10% of the baseline distribution of values for that variable (following IPCC (2014)). Extreme event probability was calculated within each pixel for each month of the year, summed over the year and then the standard deviation (SD) taken across either the model outputs or precipitation products in units of (occurrence of extreme events per year). In order to avoid spurious extremes occurring in deserts and other areas with very low variability in water cycle values, gridcells with less than 20 mm annual precipitation (multi-year mean) or <0.1 SD in their monthly precipitation across the year were excluded.

Extremes for any particular variable may only be assessed in relation to an estimate of 'normal' conditions, and for this we took a baseline distribution of values calculated at each gridcell (i.e. not globally, regionally or per biome) from an average of the five simulations involving the 2000-2013 MSWEP forcing data (Beck et al., 2017a). We took MSWEP to be our baseline product because of its high reliability and multi-source nature (satellite observations blended with reanalysis and gauge data, Beck et al. (2017a), Munier et al. (2018)) in comparison to other available products (Table 2). Carrying out the analysis on a month-by-month (e.g. comparing to a baseline calculated from all the Februaries in the MSWEP dataset) excludes spurious matching in any gridcell of e.g. winter months to summer months.

## 2.2 Uncertainty propagation

We defined three indices of uncertainty propagation $\alpha$, $\beta$ and $\varepsilon$ (Fig. 2). These indices quantify the extent to which a given simulation model increases or augments the uncertainty introduced to its simulations via the precipitation driver inputs. The $\alpha$ measure quantifies the increase or decrease in uncertainty attributable to the precipitation drivers, $\beta$ measures the equivalent for uncertainty attributable to the simulator model itself and $\varepsilon$ quantifies the overall change in uncertainty over the course of the simulation (Fig. 2). Note that the quantification of absolute uncertainty in predicted quantities (Li and Wu, 2006) is not our focus: we are instead concerned with the relative contributions of data and model uncertainty in a combination setting (Oberkampf and Roy, 2010). The defining equations are (calculated on a gridcell by gridcell basis):

$$\text{Scaled data uncertainty } \alpha_{X,j} = DOU \div DIU \tag{1}$$

$$\text{Scaled model uncertainty } \beta_{X,j} = MU \div DIU \tag{2}$$

$$\text{Scaled total uncertainty } \varepsilon_{X,j} = \alpha_{X,j} + \beta_{X,j} = (DOU + MU) \div DIU \tag{3}$$

where $DIU$ = Mean uncertainty across products in precipitation extreme occurrence (input forcing data uncertainty)

$DOU$ = Mean uncertainty across products in variable $X$ extreme occurrence (output model uncertainty attributable to forcing data input)

$MU$ = Mean uncertainty across models in variable $X$ extreme occurrence (output model uncertainty attributable to model differences)

All mean uncertainties are in units of (extreme event occurrence frequency per year: EE/yr hereafter) and $j$ can be either *high* or *low* depending on whether high or low extremes are being considered. The uncertainty propagation involves input uncertainty from the precipitation driver ($DIU$), which under the simulation is modified into the uncertainty of $X$ when averaged across the different results obtained from using different precipitation products ($DOU$), but, unlike the forcing data, the simulation results have uncertainty as a consequence of the differences between simulator model used ($MU$) which means that total uncertainty at output level is ($DOU+MU$) (Fig. 2).

In summary, $\varepsilon_{X,j}$ may be understood as a measure of how much input precipitation product data uncertainty ($DIU$) is amplified into output uncertainty ($DOU+MU$) during an ensemble of simulations. Note that it is possible for ($DOU+MU$) to be less than $DIU$ (i.e. to have $0.0 < \varepsilon_{X,j} < 1.0$), which will occur if we have models that are broadly similar in output (i.e. similar columns in the table of Fig. 2) and also little variability in the responses of those models to different levels of precipitation and/or precipitation correlates (i.e. similar rows). This may be interpreted as the ensemble models 'stabilising' the input uncertainty $DIU$ to a lower amount of uncertainty in the outputs ($DOU+MU$) and reinforces the interpretation of $\varepsilon$ as a measure of the 'augmentation' of input uncertainty as a result of model calculations.. This augmentation comes from two sources: firstly, a model ensemble can produce outputs with higher sensitivity to input precipitation e.g. through a significant nonlinear

relationship between $X$ and precipitation in the majority of ensemble models ($\alpha$), but it must not be forgotten that higher uncertainty in the outputs may also come from the differences in non-precipitation dependencies inside these models, which may also be larger in magnitude than $DIU$ ($\beta$). Division by zero in the case $DIU$=0.0 will not occur because of the masking to avoid spurious extremes in arid areas (above).

## 3 Results

Comparison of precipitation extreme event occurrences across the forcing precipitation products shows immediate differences both spatially (Fig. 3) and between the products themselves (Fig. 4). Notably, the precipitation products differ in their extreme event occurrence rates, with especially TRMM-RT presenting increased rates of extreme high precipitation events across the globe and particularly GSMaP presenting increased rates of extreme low events (for uncertainty maps, see Fig. S1, Fig. S2, Fig. S3 and Fig. S4). Calculating these absolute uncertainty values is a necessary step towards assessing the relative magnitudes of data and model uncertainty for different extreme events.

## 3.1 Scaled uncertainty

Considering firstly $\alpha_{X,j}$, the uncertainty that is directly attributable to the precipitation data products, we found that in terms of global average $\alpha_{X,j}$ was mostly <1 (i.e. $\log_{10}(\alpha_{X,j})$<0) for ET highs (58.1% *vs.* 41.9%) and decreased as precipitation increased in all latitudinal zones except the northern tropics, but for runoff highs, $\alpha_{X,j}$ increased with precipitation in all latitudinal zones except the equatorial tropics (Fig. 5). Points where data uncertainty greatly increased on propagation through models ($\alpha_{X,j}$>1) occurred mostly during the prediction of low extremes (ET or runoff) and were restricted to areas with rainfall <2000 mm/yr (Fig. 5). Points where data uncertainty greatly decreased on propagation through models ($\alpha_{X,j}$<0.1, $\log_{10}(\alpha_{X,j})$<-1) occurred mostly during the prediction of runoff extremes (mostly low extremes, but also high) and were restricted to areas with rainfall <1000 mm/yr (Fig. 5). Points with high precipitation uncertainty occurred in both dry and wet environments.

Considering $\beta_{X,j}$, the increase in model uncertainty relative to input data uncertainty, we found that $\beta_{X,j}$ was dominantly <1 (i.e. $\log_{10}(\beta_{X,j})$<0) for ET highs (80.1% *vs.* 19.8%) and decreased as precipitation increased in all latitudinal zones; for runoff highs, $\beta_{X,j}$ was also mostly <1 (55.6% *vs.* 44.4%) but increased with precipitation in all latitudinal zones except the equatorial tropics (Fig. 6).

The scaled increase in total (data + model) uncertainty is measured by $\varepsilon_{X,j}$. In all latitude zones except the northern tropics, we found that uncertainty in ET highs increased over the course of the simulation ($\varepsilon_{X,j}$ was dominantly >1 - i.e. $\log_{10}(\varepsilon_{X,j})$>0) at the great majority of locations (80.5% *vs.* 19.5%), though the magnitude of the increase reduced in wetter environments (Fig. 7). In all latitude zones except the equatorial tropics, we also found that uncertainty in runoff highs increased over the course of the simulation at the great majority of locations (76.2% *vs.* 23.8%), but for runoff the magnitude

increased with precipitation (Fig. 7). This implies that the causes of higher model uncertainty operate differentially in wet and dry environments, with dry environments being perhaps generally less well-modelled than wetter environments.

## 3.2 Global uncertainty

The global mean value of $\alpha$ is a measure of the amount a given quantity is affected as precipitation changes relative to the input precipitation data uncertainty (Eq. 1). For quantities that 'track precipitation', we would expect this to be close to 1 (e.g. runoff values, Fig. 8a), but especially in drier climates small variations in precipitation can drive much higher variation in output variables through threshold effects, so we might expect higher values in such regions (e.g. ET values, Fig. 8b).

The global mean value of $\beta_X$ is a measure of the internal model uncertainty in quantity $X$, relative to the input precipitation data uncertainty (Eq. 2), i.e. a measure of the diversity of the calculation methods used to derive $X$ between models. If quantity $X$ is equally sensitive to precipitation extremes across models, we should expect low model uncertainty and therefore low values of $\beta_X$ (e.g. under conditions where evapotranspiration and soil storage are minimal we would expect runoff highs and lows to be closely similar to precipitation highs and lows with the model introducing little modification of the input data). Our results show that evapotranspiration extremes are more sensitive to precipitation uncertainty in wet environments than dry environments (Fig. 8c).

Globally, model uncertainty was generally less than data uncertainty (Fig. 6, Fig. 8). In the equatorial tropics, ET prediction uncertainty was more attributable to data uncertainty, but runoff uncertainty was more attributable to model uncertainty, either indicating a wider variety of model representations of runoff generation processes within the tested models, or a greater dependence of ET estimates on precipitation inputs (Fig. 6).

Munier et al. (2018) found that the occurrence of flood (high runoff values) is generally more sensitive to high precipitation extremes than the occurrence of high evapotranspiration values, but that the reverse is true for low extremes. We do find this in our results as a rule of thumb across all environments (e.g. ($\varepsilon_{ET,high} < \varepsilon_{runoff,high}$) and ($\varepsilon_{ET,low} > \varepsilon_{runoff,low}$) and the same for $\alpha$ and $\beta$ in Fig. 8a), but we also note that in very dry and very wet environments this pattern does not persist (Fig. 8) and it also does not persist in all latitudinal zones when taken separately.

The total change in uncertainty over the course of the simulation of variable $X$ is measured by $\varepsilon_{X,j}$ (Eq. 3) and our values for $\varepsilon_{X,j}$ were universally >1.0, indicating that the model simulation does act effectively to increase (amplify) the uncertainty in the forcing precipitation data. This also implies that when a set of models is under consideration, model uncertainty is usually greater than data uncertainty. Finally, high uncertainty points for ET lows and runoff lows were disproportionately concentrated in the equatorial and southern tropics not only for $\varepsilon_{X,j}$ but also for both components $\alpha_{X,j}$ and $\beta_{X,j}$ (Fig. 5, Fig. 6 and Fig. 7; cf. Fig. 3).

## 4 Discussion

Model output uncertainty is always a mixture of input data uncertainty and uncertainty accumulated during the simulation (Li and Wu, 2006; Oberkampf and Roy, 2010; Van Loon, 2015). However, these uncertainties are not orthogonal in general because the models encode nonlinear relationships and therefore cannot be assumed to react consistently to different levels of precipitation input (e.g. (Bhuiyan et al., 2018a; Munier et al., 2018; Ukkola et al., 2016)). In this study we have had unprecedented access through the *eartH2Observe* project to an ensemble of simulations that has combined a selection of widely-used and validated precipitation data products with a spread of cutting edge land surface and hydrology simulation models.

### 4.1 Clear attribution of uncertainty to data and/or model sources

Under what circumstances can uncertainty in the prediction of water cycle quantities be attributed clearly to the model in use (model uncertainty) and/or to the precipitation product used to drive the model (data uncertainty)? Ukkola et al. (2016) found that land surface models diverged in evapotranspiration prediction during the dry season, and the results of our study strongly support this conclusion, with our calculated envelope of uncertainty widening in drier climates across the globe for all our uncertainty measures.

We found that high data and model uncertainty points for both ET lows and runoff lows were disproportionately concentrated in the equatorial and southern tropics. These zones are dominantly covered by tropical rainforests and savanna grasslands, so one possibility is that low fluxes in xeric environments are better characterised - both in data products and model characterisation - than low fluxes in these mesic and hydric environments. Data products are known to be more accurate away from areas with consistent cloud cover and a high occurrence of convective rainfall (Table 1) (Derin et al., 2016; Levizzani et al., 2018), which might explain this for data uncertainty, but having model uncertainty follow the same geographic distribution indicates that we must also consider uncertainties in the calculations of runoff and evapotranspiration. It seems also to be the case that the simple water balance approach taken by land surface and hydrology models becomes approximate in latitudinal zones where low flows are generally combined with higher temperatures and more episodic rainfall events (McGregor and Nieuwolt, 1998). This could indicate that using generalised approaches for all environments (e.g. the Priestley-Taylor or Penman-Monteith equations) is no longer sufficient for simulations at these spatio-temporal scales (Long et al., 2014; Wartenburger et al., 2018) or perhaps because we still lack crucial processes in these models, e.g. soil crusting or sealing, which only occur in semi-arid or arid areas (Marshall et al., 1996). However, we must also be careful to draw strong conclusions from these zones because another possibility is that this result simply confirms that these regions are where our available sources data are of lower quality (q.v. Fig. 3a).

Uncertainty in predictions of evapotranspiration lows (drought) in dry environments is especially high, indicating that these circumstances are a weak point in current modelling approaches. Importantly, our results quantify this effect and show that even though uncertainty in the precipitation inputs is highest in these environments, the uncertainty in model representation

of the processes involved is also significant and should not be ignored. A practical application of this is that when robust predictions of drought are required in very dry environments, not only should a spread of precipitation products be applied, but also more than one simulator model, and the model outputs should be validated as closely as possible against local data sources in order to ensure that conclusions drawn from these analyses are suitable for decision-making.

## 4.2 Relative importance of data and model uncertainty

When uncertainty is attributable to both model and data sources in a simulation ensemble, is data uncertainty generally the greater or the lesser? In a report for the Intergovernmental Panel on Climate Change (IPCC), Bates et al. (2008) drew attention to the high uncertainty there was in climate models in precipitation data (= *data uncertainty*), and also suggested that for aspects of the hydrological cycle such as changes in evaporation, soil moisture and runoff, the relative spread in projections (= *total uncertainty*) was similar to, or larger than, the changes in precipitation (points echoed later by Schewe et al. (2014) and others). Precipitation observations are known to have high uncertainty (Beck et al., 2017a; Bierkens, 2015; Kimani et al., 2017; Levizzani et al., 2018; Yin et al., 2015), but responses to precipitation low extremes (drought) should not be expected to be proportional to responses from the same model to precipitation high extremes (flood) (Veldkamp et al., 2018).

We found in general that the model simulations we analysed acted to augment uncertainty rather than reduce it. In percentage terms, the increase in uncertainty was most often less than the magnitude of the input data uncertainty, but uncertainty did not decrease through the model for any variable so the simulation models did not in any case act to 'stabilise' or decrease the uncertainty supplied to them through the precipitation data products used to drive them. We do agree with Wartenburger et al. (2018)'s finding that the forcing (data uncertainty) generally dominates the variance in ET extremes, but we found model uncertainty to be important in all cases analysed and very nearly the magnitude of the forcing uncertainty in both very dry and very wet environments. This is a very significant result because it implies that a focus on the reduction of both data and model uncertainty will be necessary in order to improve the prediction of water cycle extremes.

## 4.3 Sources of unquantified uncertainty

It is important to bear in mind that some sources of uncertainty exist in these water cycle quantities that are as yet unmeasured in any existing data products, and therefore cannot be analysed in this study. There is a very strong current emphasis in climate science on identifying global areas of high precipitation uncertainty, for example (Bierkens, 2015; He et al., 2017; Levizzani et al., 2018), from which we can highlight two uncertainty sources: Firstly, most precipitation products record observations of amount, not the type of precipitation (Table 2), however it is very likely that precipitation type strongly influences our precipitation data uncertainty: for example, convective processes are dominant in the precipitation generating processes in dryland ecosystems (Table 1), and different precipitation types occur at different spatial scales as well (Table 1). Secondly, our equatorial tropical zone (Fig. 1) includes the tropical rain belt (also known as the Inter-Tropical Convergence Zone, ITCZ) of low pressure, characterised by convective activity generating many storms. It is well-known that because of the transitory

nature of the cloud dynamics in the rain belt, precipitation products necessarily have higher uncertainty and, simultaneously, these conditions are of too short duration to be captured reliably in our analysis (Marthews et al., 2019).

For evapotranspiration in particular, Lopez et al. (2017) drew attention to the global lack of high quality *in situ* site data and the "inevitable scale mismatch" when using such data to calibrate Earth Observation datasets. Regional estimates of
5 evapotranspiration rely on scaling-up methods to take account of regional advection effects and, additionally, the use of estimated values for evaporation rates from unmeasured land use types. Each step in these calculations potentially introduces significant uncertainty with the result that there is currently wide variation between the values suggested by various global evapotranspiration products (Martens et al., 2017).

Finally, runoff: Surface runoff estimates are linked to precipitation and evapotranspiration estimates via the water
cycle balance equation (Beck et al., 2017b; Bierkens, 2015; Veldkamp et al., 2018). Because soil storage terms are usually taken as constant, underestimation of evapotranspiration often means overestimation of runoff and streamflow data (and *vice versa*). In this way, uncertainty in surface runoff is related to uncertainty in evapotranspiration estimates. However, because of the wide availability and high quality of global streamflow datasets (e.g. the Global Runoff Database, GRDC), and a much lower requirement for approximation and gap-filling in comparison to evapotranspiration data, runoff data is usually
considered to be of the highest quality in water balance studies.

## 4.4 Conclusions

Water resources management has become one of the most important challenges facing hydrologists and decision-makers at state and national levels, motivated by increasing water scarcity in some global regions and a higher frequency of extreme flood events in others (Bierkens, 2015; Dadson et al., 2017; Schewe et al., 2014). At the same time, precipitation extremes are
20 predicted to increase in frequency and impact under committed climate change (Ali and Mishra, 2017). Therefore, reliance on robust model predictions has never been greater (Kundzewicz and Stakhiv, 2010; Riley et al., 2017). In this study we have used an ensemble of simulation results from the *eartH2Observe* project derived from cutting-edge model simulators driven by a wide variety of precipitation observations, but the sources of uncertainty are nevertheless many and varied.

We found that models augmented uncertainty relative to the magnitude of forcing data uncertainty at the great
majority of spatial points, and therefore always did so in terms of global average uncertainty. Although, for predicting the extremes of evapotranspiration and runoff, the uncertainties inherent in the current generation of precipitation observation products are generally larger than the uncertainty introduced into the calculation by the land surface and hydrology models used, model uncertainty cannot be ignored and in many environments is comparable in magnitude to forcing data uncertainty. Therefore, in order to reduce prediction uncertainty we need very much to make progress on two fronts: (1) we need
precipitation data product uncertainty to be reduced (improved satellites are always welcome, of course, but we believe that much progress can also be made through moving towards blended products that are sensitive to more types of precipitation) and (2) we need to improve the mechanistic equations used in these models to derive water cycle quantities (including a better consideration of scale issues and domains of validity for existing equations).

It is important to resolve both data and model uncertainty much more clearly and identify exactly at which points in our linked modelling systems these uncertainties become the most significant. Our current model representation of land surface hydrological and biogeochemical processes remains approximate especially in very dry and very wet environments and there is a clear need for a better characterisation of these environmental extremes in order for us to move forward to the next

5 generation of climate and land surface prediction models.

## Acknowledgements

We gratefully acknowledge funding from the European Union Seventh Framework Programme (FP7/2007–2013) under grant agreement no. 603608, Global Earth Observation for integrated water resource assessment: *eartH2Observe*.

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

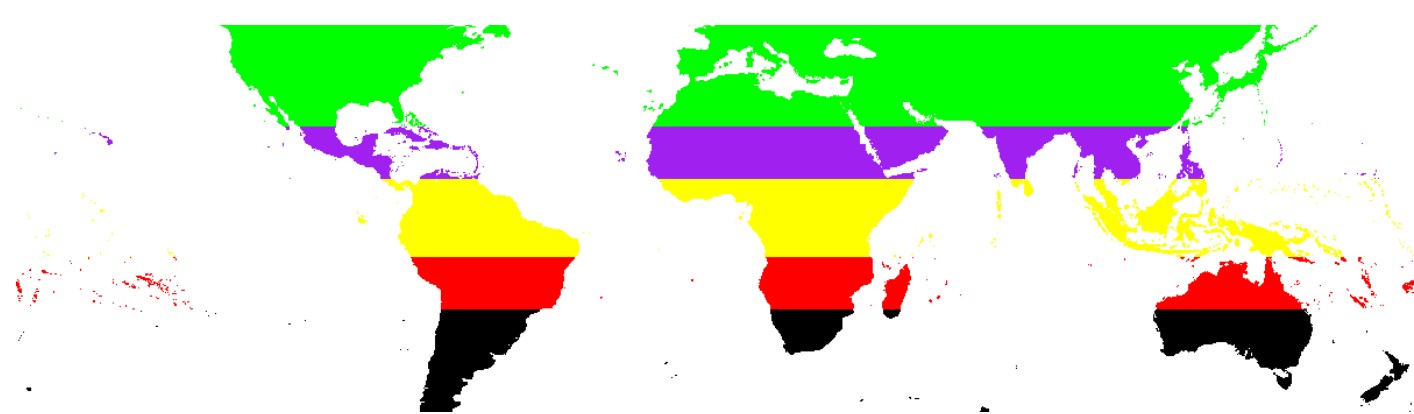

**Fig. 1**: Latitudinal zones used in this study: black = southern temperate 23.5°S to 50.0°S, red = southern
tropical 10.0°S to 23.5°S, yellow = equatorial tropical 10.0°N to 10.0°S, purple = northern tropical 23.5°N to
10.0°N and green = northern temperate 50.0°N to 23.5°S. Analyses are restricted to the area 50.0°N to
50.0°S because of the bounds of data validity in the TRMM and TRMM-RT precipitation data products (Table
2).

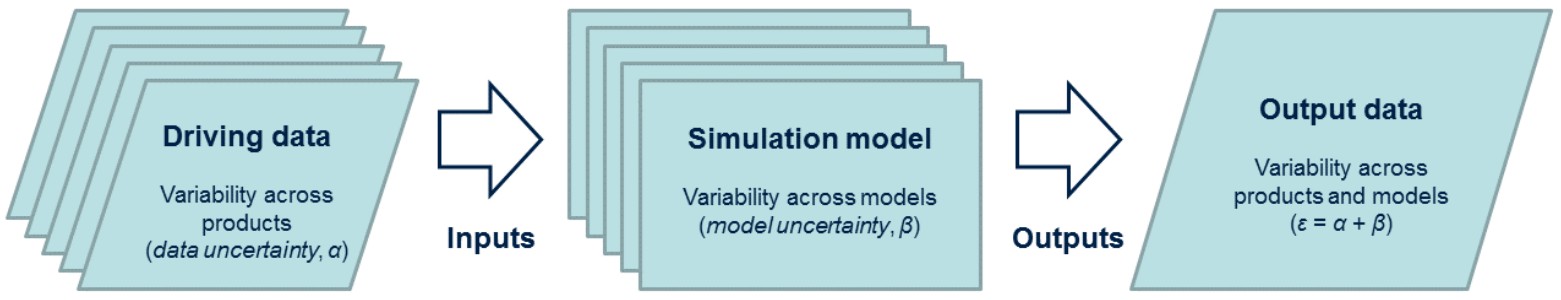

The relationship between $\alpha$, $\beta$ and $\varepsilon$ is most clearly explained by example ($P$=precipitation):

1. Say at this point and time we have 3 $P$ estimates from different data products: 5 mm, 8 mm and 10 mm. We can calculate the standard deviation $DIU$ =SD(5,8,10) =2.52 mm

2. Assume also that we have 3 models for predicting $X$=runoff:

- *Model 1* assumes runoff is equal to 2 mm/day plus an exponential contribution from $P$ if it exceeds 4 mm.
- *Model 2* is a very basic model, assuming constant runoff at this location based on the historical average, say 8.2 mm.
- *Model 3* assumes runoff is 50% of $P$ plus a contribution from groundwater return flow that ranges from 0.1 mm to 100.0 mm depending on the state of belowground aquifers.

Driving our models with those $P$ numbers to produce an estimate of $X$, we might get a table like this:

| P estimate | Runoff (mm/day) from | | | SD across models (mm/day) |
| --- | --- | --- | --- | --- |
| | *Model 1* | *Model 2* | *Model 3* | |
| 5 mm | 2.0+exp(5-4.0) = 4.7 | 8.2 | (0.50*5)+0.1 = 2.6 | 2.8 |
| 8 mm | 2.0+exp(8-4.0) = 56.6 | 8.2 | (0.50*8)+10.0 = 14.0 | 26.4 |
| 10 mm | 2.0+exp(10-4.0) = 405.4 | 8.2 | (0.50*10)+100.0 = 105.0 | 207.1 |
| SD across products (mm/day): | 217.9 | 0.0 | 56.1 | Mean from the left = 91.3 mm/day<br>Mean from above = 78.8 mm/day |

3. Note that $DOU$ = mean(SDs across products) = 91.3 mm/day, which is not equal to $MU$ = mean(SDs across models) = 78.8 mm/day (there is no constraint for these to be equal in general). We are interested in when these values are greater or less than $DIU$, so we consider the scaled uncertainties $\alpha$=($DOU \div DIU$) and $\beta$=($MU \div DIU$).

4. Note the key difference between $\alpha$, which is calculated from the outputs of the model, and $DIU$, which is calculated from the inputs: why not just consider $DIU$? Because our focus is on $X$ and therefore we need to quantify the *uncertainty introduced into X by the precipitation* ($\alpha$), which is not the same as the uncertainty in the precipitation ($DIU$) (this is an attribution study, therefore we focus on $\alpha$ rather than $DIU$).

5. In this analysis, we considered SDs of extreme event occurrence (EE/yr) rather than SDs of straight $X$ values, which we have done for two reasons: (i) this allows us to consider and compare consistently the uncertainties of different response variables with different units (e.g. $X$=runoff *vs.* $X$=evapotranspiration) and (ii) in a global analysis it is necessary to compare across biomes (e.g. a desert point with a rainforest point) and using event occurrence statistics avoids the bias towards wet or dry regions (because of their greater absolute values of e.g. runoff) that must be corrected for in studies that work with the absolute values of $X$. Using occurrence statistics doesn't change the calculations of $\alpha$, $\beta$ and $\varepsilon$ above, but does involve the additional assumption of a baseline distribution against which we may measure how extreme conditions are (see §2.1).

**Fig. 2**: Uncertainty measures quantifying how much a simulation model (land surface or hydrological model) alters the uncertainty introduced to its simulations via the precipitation driver inputs, following the *method of competing models* approach advocated for complex systems by Oberkampf and Roy (2010).

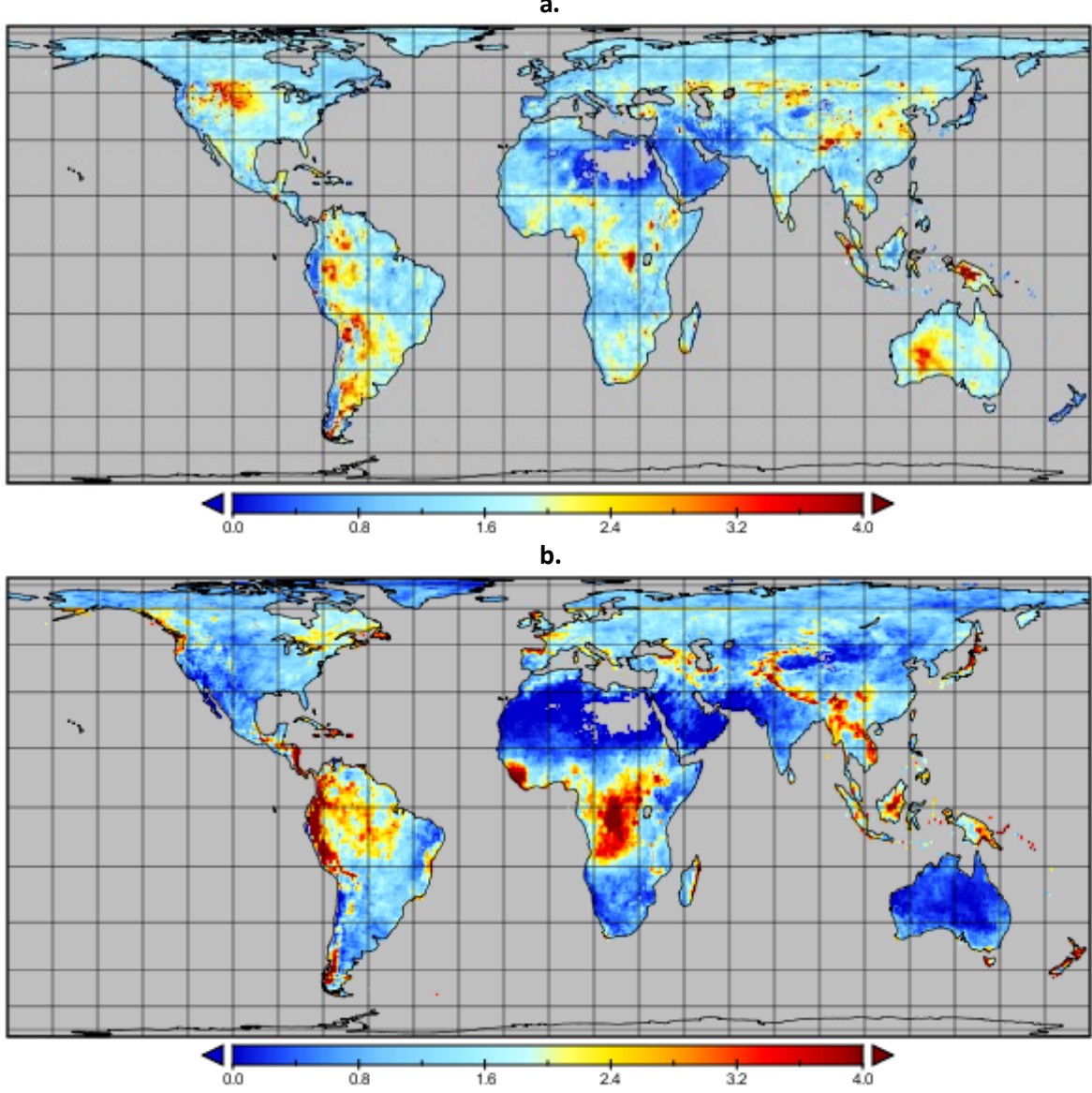

**a.**

**b.**

**Fig. 3**: Uncertainty in the precipitation inputs to the *eartH2Observe* ensemble models: (a) Uncertainty in precipitation
extreme highs and (b) Uncertainty in precipitation extreme lows (standard deviation (SD) taken across the precipitation
products) in units of (occurrence of extreme events per year). Areas of consistently very low precipitation are masked
in grey. Note that only isolated global areas exceeded 4 events/yr, so the scale is restricted to 0-4 events/yr.

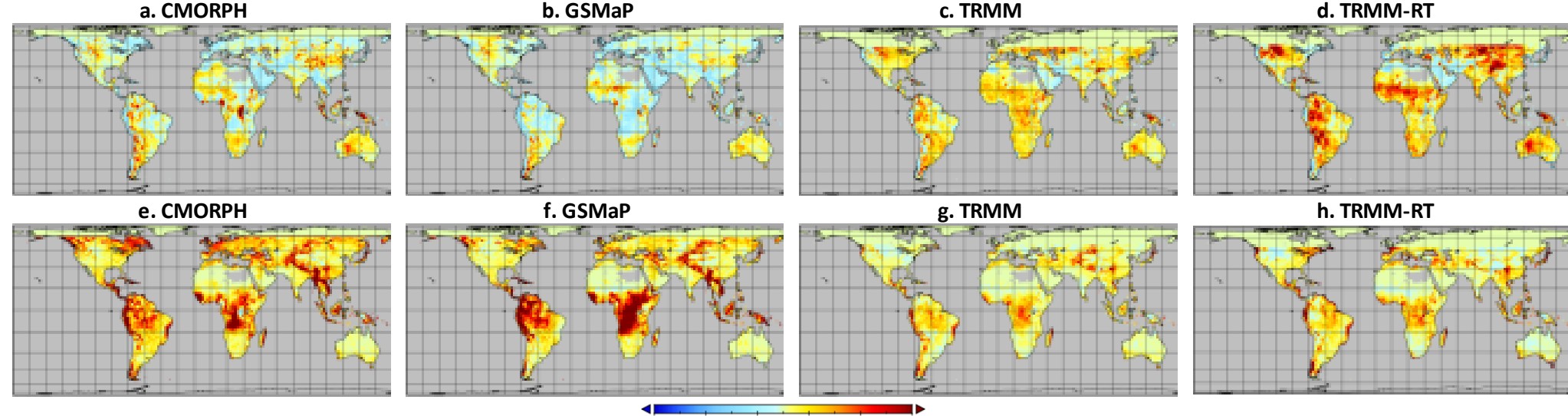

**Fig. 4**: Increase in extreme precipitation event occurrence in relation to MSWEP. Subtracting extreme high event occurrence rates in the MSWEP precipitation input from the
rates in the CMORPH precipitation input gives map (a), and (b) to (d) are the same calculation using GSMaP, TRMM and TRMM-RT instead of CMORPH. (e) to (h) is the same
calculation, but for extreme low event occurrence (i.e. the averages of the upper and lower rows are effectively the maps Fig. 3a and Fig. 3b, respectively). The clear lines at
50°N (TRMM, TRMM-RT) and 60°N (CMORPH, GSMaP) show the bounds of data validity for these products (Table 2). Note that only isolated global areas exceeded 4
events/yr, so the scale is restricted to -4 to +4 events/yr.

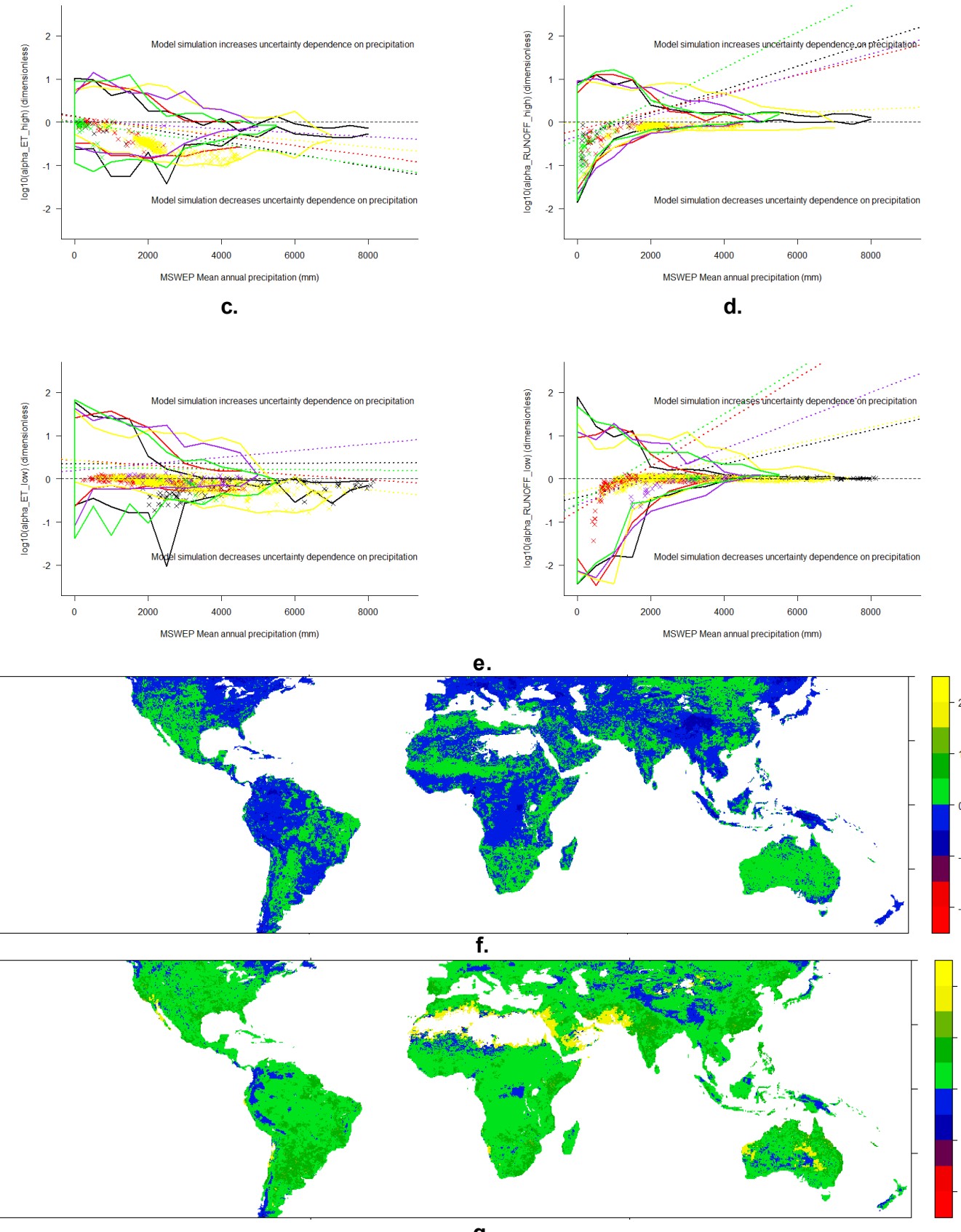

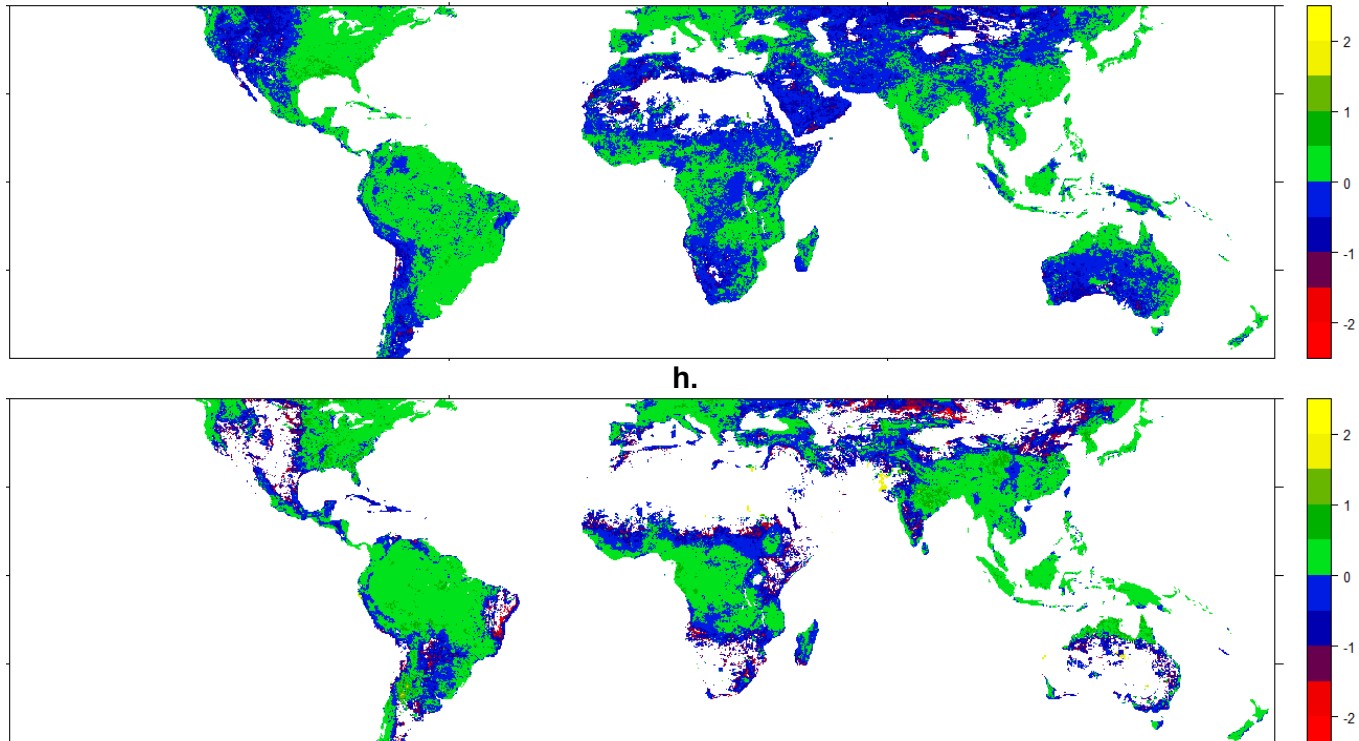

**h.**

Fig. 5: Values of $\log_{10}(\alpha_{X,j})$, where $\alpha_{X,j}$ is the scaled data uncertainty in variable $X$ (eqn 1). ($\log_{10}(\alpha_{X,j})<0$ indicates uncertainty in the predicted variable $X$ attributable to the data is less than the variability in the input precipitation forcing data; >0 indicates uncertainty in the predicted variable $X$ is greater), where $X$ is evapotranspiration (a, c, e, f) or runoff (b, d, g, h) and $j$ refers to either high extremes (a, b, e, g) or low extremes (c, d, f, h). Points on the scatter plots are coloured according to latitudinal zones (Fig. 1). Because of the density of overlapping points, only the envelope of points for each latitudinal zone is shown and the points with the highest uncertainty (uncertainty $DIU \geq (2/3)^*$(global maximum of $DIU$) ). Linear regression lines for each latitudinal zone indicate the trend as precipitation increases within each zone (all regressions were significant at the 1% level), although n.b. we do not contend in any way that the distribution of points shown is linear: these lines simply indicate a trend that is not clear to the eye from the envelopes displayed (which do not show the complete point cloud). Maps (e-h) show the corresponding spatial distributions of $\log_{10}(\alpha_{X,j})$ values for each variable, with the colour scales corresponding to the vertical axis on scatter plot (a).

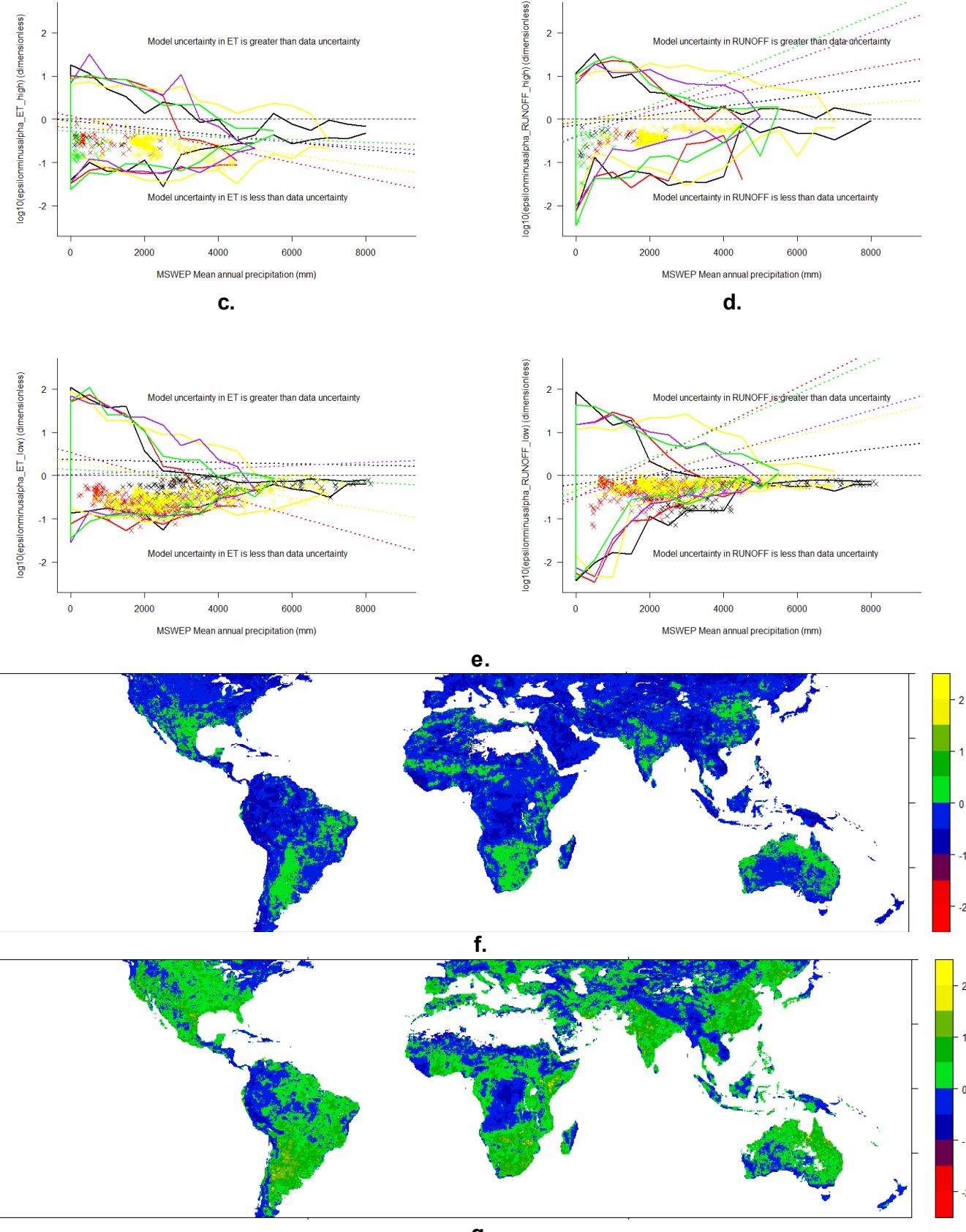

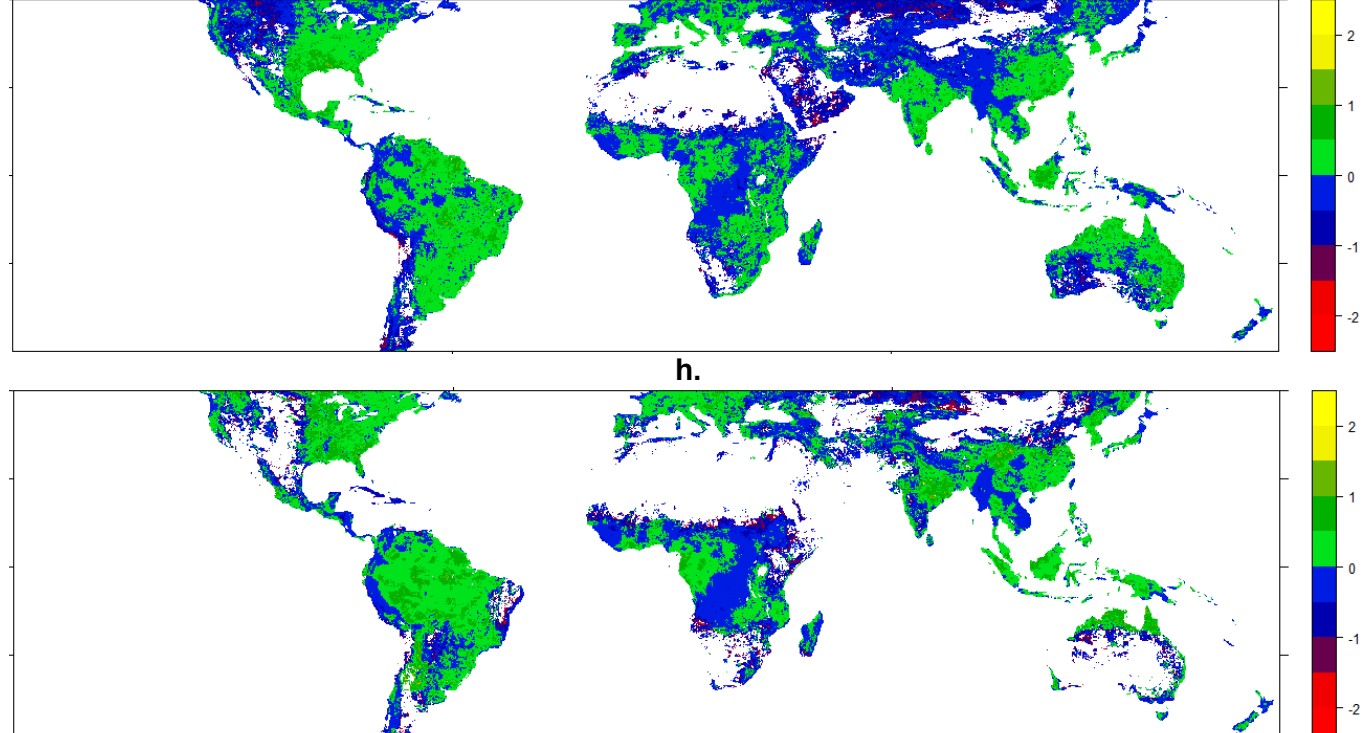

**h.**

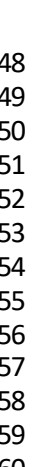

**Fig. 6**: Values of $\log_{10}(\beta_{X,j})$, where $\beta_{X,j}$ is the scaled model uncertainty in variable $X$ (eqn 2). ($\log_{10}(\beta_{X,j})<0$ indicates model uncertainty in the predicted variable $X$ is less than the variability in the input precipitation forcing data; $>0$ indicates model uncertainty in the predicted variable $X$ is greater), where $X$ is evapotranspiration (a, c, e, f) or runoff (b, d, g, h) and $j$ refers to either high extremes (a, b, e, g) or low extremes (c, d, f, h). Points on the scatter plots are coloured according to latitudinal zones (Fig. 1). Because of the density of overlapping points, only the envelope of points for each latitudinal zone is shown and the points with the highest uncertainty (uncertainty $DIU \geq (2/3)$*(global maximum of $DIU$) ). Linear regression lines for each latitudinal zone indicate the trend as precipitation increases within each zone (all regressions were significant at the 1% level), although n.b. we do not contend in any way that the distribution of points shown is linear: these lines simply indicate a trend that is not clear to the eye from the envelopes displayed (which do not show the complete point cloud). Maps (e-h) show the corresponding spatial distributions of $\log_{10}(\beta_{X,j})$ values for each variable, with the colour scales corresponding to the vertical axis on scatter plot (a).

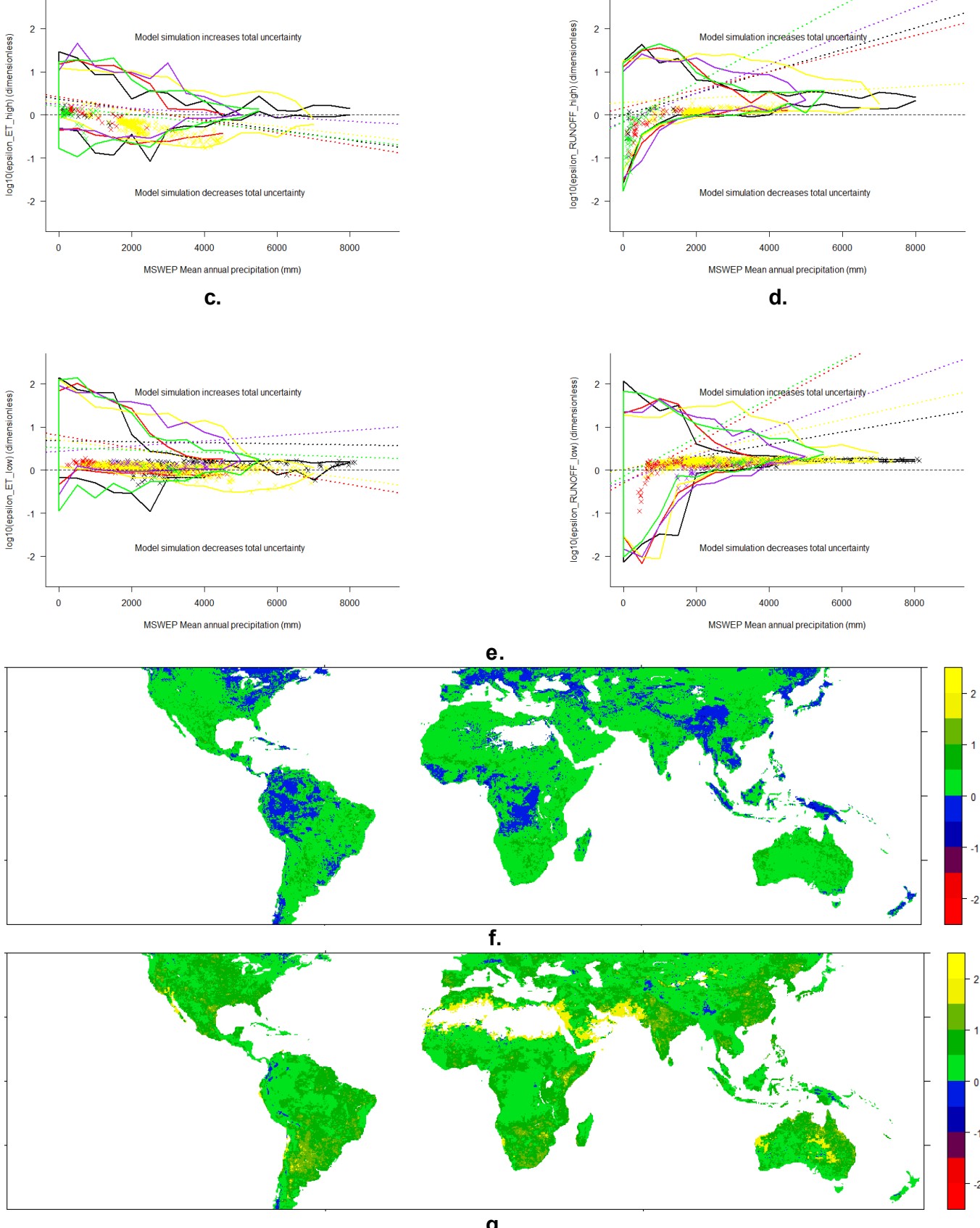

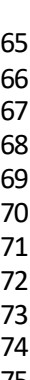
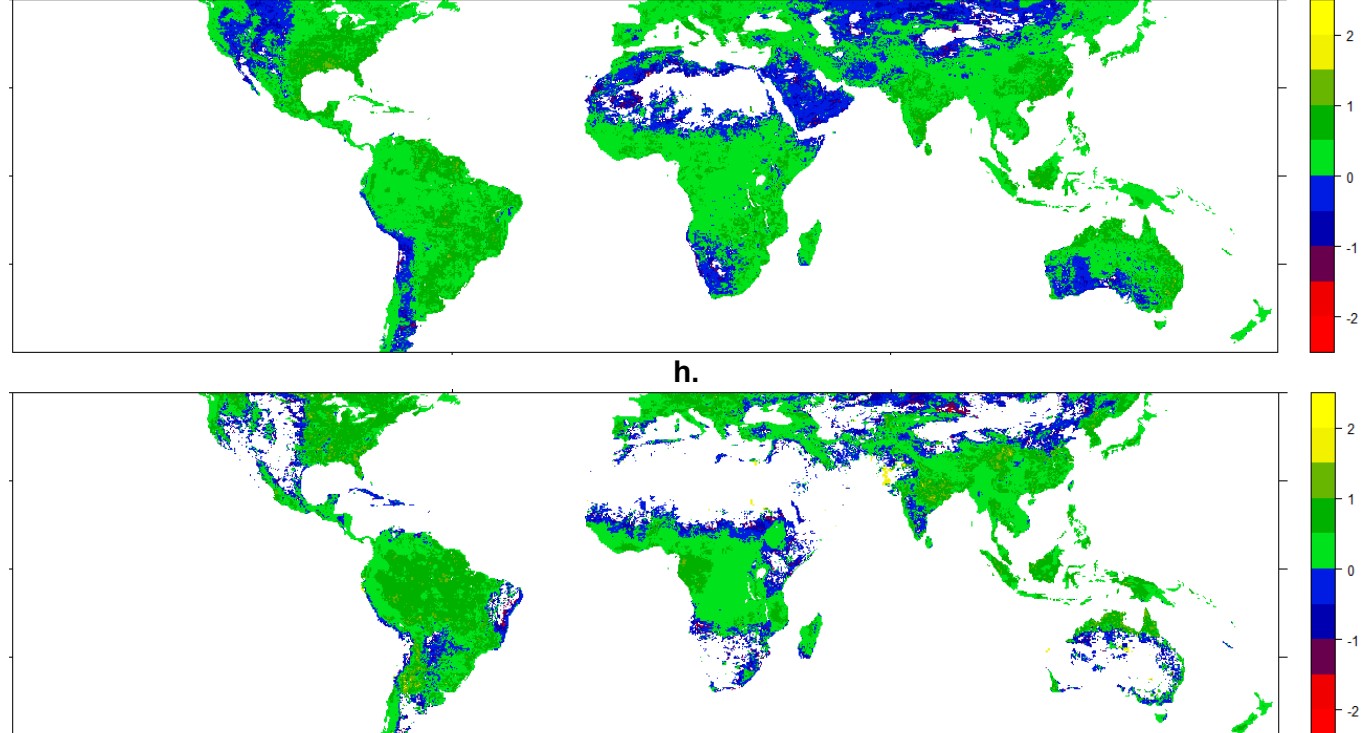

**h.**

**Fig. 7**: Values of $\log_{10}(\varepsilon_{X,j})$, where $\varepsilon_{X,j}$ is the total uncertainty in variable $X$ (eqn 3), where $X$ is evapotranspiration (a, c, e, f) or runoff (b, d, g, h) and $j$ refers to either high extremes (a, b, e, g) or low extremes (c, d, f, h). Points on the scatter plots are coloured according to latitudinal zones (Fig. 1). Because of the density of overlapping points, only the envelope of points for each latitudinal zone is shown and the points with the highest uncertainty (uncertainty $DIU \geq (2/3)^*$(global maximum of $DIU$) ). Linear regression lines for each latitudinal zone indicate the trend as precipitation increases within each zone (all regressions were significant at the 1% level), although n.b. we do not contend in any way that the distribution of points shown is linear: these lines simply indicate a trend that is not clear to the eye from the envelopes displayed (which do not show the complete point cloud). Maps (e-h) show the corresponding spatial distributions of $\log_{10}(\varepsilon_{X,j})$ values for each variable, with the colour scales corresponding to the vertical axis on scatter plot (a).

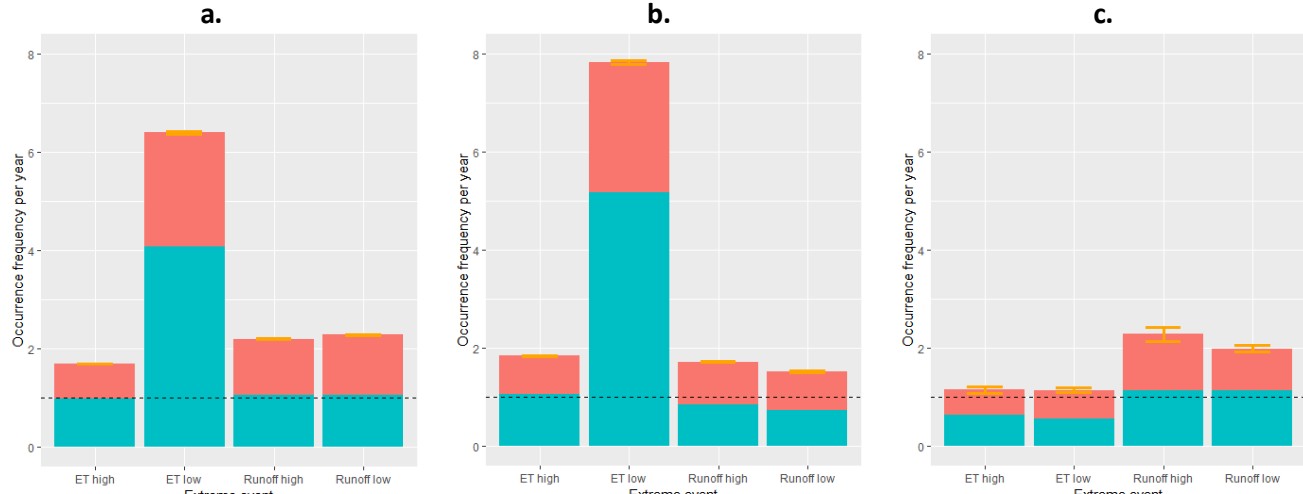

**Fig. 8**: Global mean values (averaged over 50°S to 50°N) from scatter plots in Fig. 5, Fig. 6 and Fig. 7. Plots show (a) all values, (b) values from dry environments with mean annual precipitation <1000 mm/yr only and (c) values from wet environments ≥6000 mm/yr only. Bar heights are $\varepsilon$ values (scaled total uncertainty), with ■ showing $\alpha$ values (scaled data uncertainty) and ■ $\beta$ (scaled model uncertainty); error bars show SE.

Oberkampf, W. L., and C. J. Roy (2010). *Verification and Validation in Scientific Computing*, Cambridge University Press.

3 **Table 1**: Types of precipitation and their main controlling factors (McGregor and Nieuwolt, 1998).

| Precipitation type | Spatial scale | Characteristics | Challenges |
|---|---|---|---|
| Cyclonic (=frontal) | Synoptic, regional | The leading edge of a warm and moist air mass (warm front) meets a cool, dry air mass (cold front). The warmer air mass rises over the cooler air, with precipitation occurring along the front. If the air begins to circulate, a cyclonic storm can occur. | • It is widely accepted that global warming will lead to a higher water-holding capacity for the atmosphere as well as increased rates of evaporation, and therefore increased extreme weather (Trenberth et al., 2015; Yi et al., 2015). However, the mechanisms through which the location and magnitude of these extreme events may be predicted (e.g. tipping points, thresholds) remain inadequately understood (Marthews et al., 2012). |
| Orographic | Intermediate | Warm, moist air entering a mountain range is forced to rise, and then cools and precipitation ensues (= *orographic lift*). | • Scale is an important issue: mountains can modify large-scale circulation, causing changes in local moisture convergence, but local condensation and microphysical processes also influence flow stability upstream (Marthews et al., 2012). |
| Convective | Local (often sub-grid) | A warm soil or vegetation surface warms the air above it, which then rises vertically and cools, with precipitation occurring on cooling.<br><br>'Convection-permitting' model runs usually require a sub-daily timestep and <10 km spatial resolution, and in the absence of these a convection parameterisation scheme (CPS) is necessary (i.e. assumptions about subgrid and subdaily dynamics) (Prein *et al.* 2015). | • *Stratiform precipitation* is when the rise is diagonal rather than vertical (i.e. similar to orographic, but not as a result of landform)<br>• Sub-grid displacement of cloud occurrence from driver (Taylor et al., 2012)<br>• Land surface exchange (e.g. evapotranspiration) has a significant effect, but often not modelled explicitly.<br>• Resolution of snow versus rainfall in mountain regions is critical for water resources management, but not well-characterised in models.<br>• CPSs generally overestimate light rain (drizzle) because they overestimate the number of precipitation days (by equating clouds with rain) and / or underestimate precipitation intensity (Marthews et al., 2012; Prein et al., 2015). Conversely, it is a known limitation of some satellites that they are not sensitive to, and therefore underestimate, light rain (e.g. Luo et al. (2017)). This introduces a 'calibration gap': calibration of large-scale models against satellite-based precipitation observations must not only factor out the overestimation of CPSs, but also the underestimation of the observations. |

7
8 **Table 2**: Global precipitation products used to drive the models selected from Dorigo *et al.* (2014). Data files used are
9 available through the Water Cycle Integrator https://wci.eartH2Observe.eu/ at 25 km resolution for the period 2000-
10 2013. Algorithm type is as given by the International Precipitation Working Group (IPWG) [*].

| Product | Algorithm | Notes |
|---|---|---|
| **Multi-Source Weighted-Ensemble Precipitation (MSWEP)** | | Global reanalysis data (Beck et al., 2017) |
| **Climate Prediction Center MORPHing Technique (CMORPH)** | Blended microwave-infrared | Restricted to 60°S to 60°N<br><br>A passive microwave-based product advected in time using geosynchronous infrared data (Joyce et al., 2004). When microwave observations are not available, infrared observations are used to advect the last microwave scan over time. In addition to advecting precipitation forward in time, the algorithm propagates precipitation backward once the next microwave observation becomes available (Mehran and AghaKouchak, 2014). |
| **Global Satellite Mapping of Precipitation (GSMaP)** | Blended microwave-infrared | Restricted to 60°S to 60°N (Tian et al., 2010) |
| **Tropical Rainfall Measuring Mission (TRMM)** | Satellite-based | Restricted to 50°S to 50°N |
| **TRMM Real Time (TRMM-RT)** | Satellite-based | Restricted to 50°S to 50°N<br><br>Mainly based on microwave data aboard Low Earth Orbit satellites (Huffman et al., 2007). The TRMM-RT algorithm is primarily based on microwave observations from low orbiter satellites. Gaps in microwave observations are filled with infrared data (Mehran and AghaKouchak, 2014). |

[*] *Real-time* usually = there is at most a 1-2 hour delay before observation data is made available raw (i.e. with no gap-
filling or other modification).
*Near-real-time* = there is at most a 1-2 day delay before delivery, allowing some initial data checks to be carried out.
*Reanalysis data* = data assimilation techniques have been used to fill gaps in the observation data (e.g. missing
variables).
*Blended* = observation data have been combined with either or both of raingauge and reanalysis data to create a more
robust and quality-controlled product.

**Table 3**: Modelling systems details (Dutra et al., 2015; Nikolopoulos et al., 2016). Each model was driven using as close as possible to the same configuration: Global Water Resources Reanalysis 2 (WRR2, Arduini et al. (2017) and http://jules.jchmr.org/content/research-community-configurations). Simulation results are available on the THREDDS data server (https://wci.eartH2Observe.eu/thredds/catalog.html, see Schellekens et al. (2017)).

| Model | Institution | Simulations |
|---|---|---|
| Hydrology Tiled ECMWF Scheme for Surface Exchanges over Land model (**H-TESSEL**) (Balsamo et al., 2009) | ECMWF | A 10-year spin-up was carried out: an initial run from 1 January 1979 to 1 January 1989, while the land surface state of January 1989 was used to initialize the main simulation. |
| JULES is the Joint UK Land Environment Simulator model (**JULES**) (Best et al., 2011; Clark et al., 2011) | MetO/CEH | A 10-year spin-up was carried out: an initial run from 1 January 1979 to 1 January 1989, while the land surface state of January 1989 was used to initialize the main simulation. |
| ORganizing Carbon and Hydrology In Dynamic EcosystEms model (**ORCHIDEE**) (d'Orgeval et al., 2008; Krinner et al., 2005) | CNRS/IPSL | The model was spun up with a simulation from 1 January 1979 to 31 December 1990. This simulation started with an average soil moisture and empty aquifers. After the 12 years of spin-up, river discharges have reached equilibrium. |
| SURFace EXternalisée model (**SURFEX**) (Decharme et al., 2011; Decharme et al., 2013) | Météo-France | A 20-year spin-up was carried out using the 1979–1988 period twice. |
| Water – Global Assessment and Prognosis-3 (**WaterGAP3**) (Schneider et al., 2011; Verzano et al., 2012). A grid-based, integrative global fresh water resource assessment tool. | University of Kassel | Storage compartments were initialized by re-running the model with the first year of available meteorological forcing 10 times.<br><br>WaterGAP includes a water use model (domestic and industrial water use are parameterised as a function of average income per country (GDP/capita), allowing global water use calculations. |

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
