# Peer review of "A global scale evaluation of extreme event uncertainty in the *earth2Observe* project"

_Hydrology and Earth System Sciences, 2018_

## Referee Comment (RC1) · Anonymous Referee #1 · 25 Feb 2019

General comments The manuscript presents an analysis of a unique dataset that was produced from the eartH2Observe project. This dataset involves the simulation of several hydrologic variables from a number of state-of-art land surface/hydrologic models and using as forcing several satellite and reanalysis dataset. The scale of analysis is global and the focus is on the tails (i.e. low/high extremes) of evapotranspiration and surface runoff. Overall the work is very interesting and the dataset analyzed is very unique. Additionally, the fact that the analysis is performed at global scale provides important information on the regional variability of findings. The manuscript is generally well written but there are certain parts (especially in the description of methodology) that require additional clarification and discussion. I provide some specific comments below that hopefully will help the authors to improve their manuscript.

[Figure]

Specific comments

1. I believe that the title should be revised to better reflect the context of the paper. One of the main elements of you analysis is "uncertainty in identification of extreme events" but this is not reflected in the current title.

2. Abstract L17-19: I agree but given the focus of your analysis (i.e. identification of extremes) you should be more specific on what your results will allow to comment. For example, models can be quite robust in representing the main body of the distribution of hydrologic variables, which is actually very important for water resources applications. So I suggest to specifically refer again to representation of extremes.

3. P3L3: For a multiregional evaluation of satellite precipitation over complex terrain, you may want to consider also Derin, Yagmur, et al. "Multiregional satellite precipitation products evaluation over complex terrain." Journal of Hydrometeorology 17.6 (2016): 1817-1836.

4. Information in Section 2.1 needs to further clarified. What do you mean by "base distribution"? Is this the reference for your EE/yr at each cell? Why you average the five runs and you don't consider each model independently? Do you repeat the same procedure for each product and then compare? Please clarify.

5. It would be very useful to provide a graphical example to demonstrate the different uncertainty components that you describe in equations 1-3.

6. L24-28 are confusing. First, it is not clear why you consider $\varepsilon x,j > 1$ as an indicator of model amplification of uncertainty? Do you mean $\alpha x,j$ instead? Also if you want to identify the relative contribution of the different sources of uncertainty, why don't you take the ratio of $\alpha/\beta$ ?

7. P6L6: "global average", why do you consider global average? It is not advisable since the average masks regional variability. Also "ET highs (58.1% vs 41.9%)", it is not clear what these numbers correspond to.
8. P6L10 "$\alpha$x,j<-1", I believe you mean log($\alpha$x,j).

9. P6L19-23. Interesting findings, some additional comments are welcome here. For example, why "the magnitude of the increase reduced in wetter environments"?

10. P6L25: "The global mean value….is a measure of variability". How can a mean value tell you anything about variability? Please clarify/revise.

11. P6L25-30: In general, this part of the text is quite difficult to "digest". Please improve clarity.

12. P6L31: What do you mean by "internal model uncertainty"?

13. P7L3-4: "…are more sensitive to precipitation extremes in wet environments". Be careful here, you should state "…more sensitive to precipitation uncertainty".

14. P7L15-16: I believe that there is a confusion here between model uncertainty and uncertainty propagation. This is a very important aspect and the authors should clarify it in their discussion. For example, even with zero model uncertainty, transformation of precipitation uncertainty to runoff uncertainty could potentially amplify as a result of the nonlinear transformation of rainfall-to-runoff.

15. The same point as in 14(above) should be considered in the discussion of section 4.2 (e.g. L26-27).

16. P9L10: "…to improve prediction of water cycle quantities". Ok I agree but the analysis presented has not done anything on the quantitative aspect. Perhaps revise to "improve prediction of water cycle extremes"?

17. Section 4.3. (L15-22). The text here is relevant to work that is evaluating uncertainty and compares against observations. However, this is not the scope of your work. You isolate (correctly) the forcing and model uncertainty by considering as reference a model/forcing combination.

18. Fig2: What is (a) and what is (b). Also, some of the explanation on the calculation

of results could be added to text in manuscript as well.

19. Fig3. Similar comment on the explanation.

20. Figure 4. I find this map very useful. It would be nice to provide for the other cases analyzed.

21. Figure 7: "erros bars show SE". Do you mean standard error? And how the error is defined. Perhaps you refer to standard deviation instead?

―――――――――――――――――

---

## Author Comment (AC1) · 13 Mar 2019

Response to Anonymous Referee #1 ():

General comments The manuscript presents an analysis of a unique dataset that was produced from the eartH2Observe project. This dataset involves the simulation of several hydrologic variables from a number of state-of-art land surface/hydrologic models and using as forcing several satellite and reanalysis dataset. The scale of analysis is global and the focus is on the tails (i.e. low/high extremes) of evapotranspiration and surface runoff. Overall the work is very interesting and the dataset analyzed is very unique. Additionally, the fact that the analysis is performed at global scale provides important information on the regional variability of findings. The manuscript is generally

well written but there are certain parts (especially in the description of methodology) that require additional clarification and discussion. I provide some specific comments below that hopefully will help the authors to improve their manuscript.

– RESPONSE – Thank you very much for the review and we hope that the responses below are sufficient reply to these useful comments.

Specific comments 1. I believe that the title should be revised to better reflect the context of the paper. One of the main elements of you analysis is "uncertainty in iden­tification of extreme events" but this is not reflected in the current title. – RESPONSE – We agree and have added "uncertainty" to the title.

2. Abstract L17-19: I agree but given the focus of your analysis (i.e. identification of extremes) you should be more specific on what your results will allow to comment. For example, models can be quite robust in representing the main body of the distribution of hydrologic variables, which is actually very important for water resources applications. So I suggest to specifically refer again to representation of extremes. – RESPONSE – We agree: we have now specified in the abstract that "Our results are important for highlighting the relative robustness of satellite products in the context of land surface simulations of extreme events" as requested.

3. P3L3: For a multiregional evaluation of satellite precipitation over complex terrain, you may want to consider also Derin, Yagmur, et al. "Multiregional satellite precipitation products evaluation over complex terrain." Journal of Hydrometeorology 17.6 (2016): 1817-1836. – RESPONSE – Thank you for this reference: it was indeed relevant and we have used it at two points in the revised text.

4. Information in Section 2.1 needs to further clarified. What do you mean by "base distribution"? Is this the reference for your EE/yr at each cell? Why you average the five runs and you don't consider each model independently? Do you repeat the same procedure for each product and then compare? Please clarify. – RESPONSE – Thank you for pointing this out: we had not explained why a base distribution (or baseline

distribution) is necessary. We have added to section 2.1 "Extremes for any particular variable may only be assessed in relation to an estimate of 'normal' conditions, and for this we took a baseline distribution of values ...". At this point we have not considered each model independently because we do that at a later point when comparing model uncertainty and data product uncertainty. Finally, yes we did exactly repeat the same procedure for each product (and each model) and compare: this is the basis of the definitions of alpha, epsilon and beta later on in the method.

5. It would be very useful to provide a graphical example to demonstrate the different uncertainty components that you describe in equations 1-3. – RESPONSE – We have indeed included a graphical example of the different uncertainty components: this is Fig. 7. We have considered moving that figure to an earlier point in the paper, but it is difficult to do so because we need to explain more details of how it is calculated before presenting that figure.

6. L24-28 are confusing. First, it is not clear why you consider $\varepsilon$x,j >1 as an indicator of model amplification of uncertainty? Do you mean $\alpha$x,j instead? Also if you want to identify the relative contribution of the different sources of uncertainty, why don't you take the ratio of $\alpha/\beta$ ? – RESPONSE – We apologise but we have to disagree with the reviewer on this point and stand by the text as we have written it: it is indeed epsilon rather than alpha that measures model amplification. Take this example: Let's say at a particular dryland point on the globe precipitation varies over the range 400-1000 mm/yr. According to our definitions, which are phrased in terms of extreme event occurrence, data input uncertainty DIU will perhaps be 2.5 EE/yr (say we are concentrating on high extremes and those precip numbers equate to extreme high precip occurring between 2 and 7 times per year). Say model 1 is quite 'flashy' in the sense that runoff is very sensitive to precip but ET is calculated from an empirical relationship quite insensitive to precip extremes, DOU might be high for runoff highs (e.g. 5.0 EE/yr, say) but very low for ET highs (e.g. 1.5 EE/yr). For model 2 you might easily have the reverse with runoff DOU being low and ET DOU being high. Across all models, let's say

runoff is always flashy; whereas ET is sometimes flashy and sometimes not, depending on whether a conservative empirical representation has been used in that model or something more realistic. With these example values, we would expect to see alpha always high for the 'flashy' runoff and quite high for ET too (the flashy runs averaging out the nonflashy ones). We would also expect beta to be low for runoff (because it is consistent across models) but high for ET. Epsilon=alpha+beta is a measure of the total propagation of uncertainty and this is the most appropriate measure to look at when talking about uncertainty propagation, because to look at only alpha is equivalent to ignoring the model uncertainty, and this is one of the important points we are trying to make with this paper. To look at the ratio of alpha to beta is also not appropriate because for that DIU drops out of the calculation and we cannot take account of the variation in the driving data (see eqn1 and eqn2). We have added some text to section 2.2 ("In summary ... product only") that we hope clarifies this point and the relationship between alpha, epsilon and beta.

7. P6L6: "global average", why do you consider global average? It is not advisable since the average masks regional variability. Also "ET highs (58.1% vs 41.9%)", it is not clear what these numbers correspond to. – RESPONSE – This is an introductory point at the start of section 3.1 which we then expand upon in more detail later. It is not irrelevant to point out that the alpha values are universally quite small and do seem to decline with increasing precipitation. the regional variability is displayed graphically in Figs. 4-6.

8. P6L10 "$\alpha$x,j<-1", I believe you mean log($\alpha$x,j). – RESPONSE – This statement is correct as stated (a log cannot be negative and we felt it was clearer to quote the bound in terms of straight alpha at this point).

9. P6L19-23. Interesting findings, some additional comments are welcome here. For example, why "the magnitude of the increase reduced in wetter environments"? – RESPONSE – We feel that it would be too speculative to include here any of the various theories that could explain why the magnitude of the increase is reduced in wetter

environments. For example, there could be a saturation effect in the environment (but in the absence of soils or land use data we cannot be sure of this) or fast drainage could occur more often under more episodic rainfall (but we have no data on drainage patterns) or the occurrence of convective cells might be very regionally specific (but these are not even visible on most remote sensing products). We have tried to be careful to stick to discussing points that are directly relevant to the results and data that we have presented and we hope in a later study to look at these trends in more detail, but we have omitted any discussion of this here.

10. P6L25: "The global mean value. . ..is a measure of variability". How can a mean value tell you anything about variability? Please clarify/revise. – RESPONSE – If the quantity in question (alpha) is itself a measure of variability, then the mean of alpha will still be a measure of variability even though we agree it will not contain any information about the variability of alpha itself. We have revised the wording here to avoid the apparent contradiction.

11. P6L25-30: In general, this part of the text is quite difficult to "digest". Please improve clarity. – RESPONSE – We agree and thanks: we have removed the middle sentence, which we hope has improved the clarity of the paragraph.

12. P6L31: What do you mean by "internal model uncertainty"? – RESPONSE – We have added in the explanation that this is "a measure of the diversity of the calculation methods used to derive X between models".

13. P7L3-4: ". . .are more sensitive to precipitation extremes in wet environments". Be careful here, you should state ". . .more sensitive to precipitation uncertainty". – RESPONSE – Corrected with thanks.

14. P7L15-16: I believe that there is a confusion here between model uncertainty and uncertainty propagation. This is a very important aspect and the authors should clarify it in their discussion. For example, even with zero model uncertainty, transformation of precipitation uncertainty to runoff uncertainty could potentially amplify as a result of

the nonlinear transformation of rainfall-to-runoff. – RESPONSE – We have specifically defined separate quantities for model uncertainty (beta) and uncertainty propagation (epsilon) and we believe that we have not confused the two issues in this paper: in fact, drawing attention to the difference is one of the overall points of the paper. If runoff is generally 1000-1500 mm/yr with 7 peaks/yr when precipitation inputs are 500-1000 mm/yr with 3 peaks/year, then output uncertainty differs not only in terms of absolute value (which can be a linear effect) but also in terms of distribution (a nonlinear effect). By focusing our study on extreme event occurrence, linear effects should be cancelled out (as long as the extremes are calculated in terms of an appropriate baseline for each quantity, which we have done), however of course there will be nonlinear effects that can give nontrivial values to epsilon (and alpha) even in the case of beta=0 because the number of peaks may still change. At no point in the paper have we assumed that this will not happen: in fact, we have accounted for this in all analyses. At P7L15-16 we have simply stated that model uncertainty is usually greater than data uncertainty. We believe that the reviewer here does not like the implication that when model uncertainty is small then data uncertainty must be even smaller, and it was certainly not our intention to imply that. We have modified the text to say "when a set of models is under consideration, model uncertainty is usually greater than data uncertainty". To avoid the same implication we have added "in a simulation ensemble" to the start of section 4.2 as well.

15. The same point as in 14(above) should be considered in the discussion of section 4.2 (e.g. L26-27). – RESPONSE – Please see last point #14.

16. P9L10: ". . .to improve prediction of water cycle quantities". Ok I agree but the analysis presented has not done anything on the quantitative aspect. Perhaps revise to "improve prediction of water cycle extremes"? – RESPONSE – Thank you for the suggestion: changed to "extremes"

17. Section 4.3. (L15-22). The text here is relevant to work that is evaluating uncertainty and compares against observations. However, this is not the scope of your work.

You isolate (correctly) the forcing and model uncertainty by considering as reference a model/forcing combination. – RESPONSE – These comments are made in a section entitled "4.3 Sources of unquantified uncertainty" and we state clearly in the preceding sentence that we could not analyse these kind of situations in our particular study given the data available. However, we find these issues the be entirely within scope of this study and, in fact, we would have been remiss not to have mentioned them. We hope very much in a follow-up study to find some way to tackle these sorts of issues and we believe it is entirely appropriate to have a brief mention of them here.

18. Fig2: What is (a) and what is (b). Also, some of the explanation on the calculation of results could be added to text in manuscript as well. – RESPONSE – The legend stated "a. Uncertainty in precipitation extreme highs and b. Uncertainty in precipitation extreme lows", which perhaps was not clear because we did not use parentheses on the (a) and (b), so parentheses have been added in. We agree that the explanatory text in the legend was perhaps too long and was mostly superfluous because the calculation is already described in the main text (section 2.1) so we have now omitted it.

19. Fig3. Similar comment on the explanation. – RESPONSE – Unlike for Fig. 2, the description of the calculation for Fig. 3 is not repeated in the main text, but after reconsidering the legend we would like to argue that the amount of detail here is appropriate: the explanation here simply describes what the rows and columns of this multi-panel plot display and we do not see any way to appreviate this without forcing the reader to hunt through the text for this description. Therefore we have left this text as it is and we hope the reviewer will either reonsider this comment or specify more precisely what change he/she would like us to make, please?.

20. Figure 4. I find this map very useful. It would be nice to provide for the other cases analyzed. – RESPONSE – The other 3*3=9 maps from Figs. 4, 5 and 6 were excluded before simply from space considerations. They have now been added.

[Figure]

21. Figure 7: "erros bars show SE". Do you mean standard error? And how the error is defined. Perhaps you refer to standard deviation instead? – RESPONSE – We have left this text as it is: what was calculated here was standard error, which differs from standard deviation because you divide by the square root of sample size (the abbreviation SE is standard). The "averaged over 50°S to 50°N" earlier in the legend makes it clear that this is calculated across gridcells rather than time (i.e. sample size is the number of gridcells in this case).

---

## Author Comment (AC2) · 14 Mar 2019

The reviewer stated: "8. P6L10 "$\alpha$x,j<-1", I believe you mean log($\alpha$x,j)."

I responded that the statement at P6L10 was correct as stated, however I need to apologise for that (!): actually the reviewer was 100% correct on this point. I have to admit I got my log values confused at this point and didn't even realise it when I initially read this reviewer comment. I retract completely: the statement was not correct: the reviewer was! The text has now been amended to "$\alpha$X,j<0.1, log10($\alpha$X,j)<-1".

Best, Toby M
* * *
[Figure]

622, 2019.

---

## Referee Comment (RC2) · Anonymous Referee #2 · 27 Mar 2019

Review of Marthews et al. "A global scale evaluation of extreme events in the earth2observe project"

The authors use model simulations from the earth2observe project to study the sources of uncertainty in simulated runoff and evapotranspiration (ET). Model simulations from this project are well chosen for this purpose as they are performed with (i) different precipitation forcing datasets and (ii) different land surface and hydrological models. Analysing these simulations, the authors compared the relative importance of the precipitation forcing uncertainty with that of the model uncertainty for resulting runoff and ET extremes.

———————————————

Recommendation: I think the paper should be rejected.

While the research question is interesting and relevant, and the model simulations are well suited for the purpose of this study, the applied methodology is too complex and hard to understand such that I am not sure about the robustness of the resulting conclusions.
* * *
General comments:

(1) As mentioned above I do not understand (the purpose of) the methodology applied in this study, even after carefully reading it many times.

While the focus on extremes is not explained or motivated, I also do not see why/how 10% of a 14-year time series can already be considered extreme. Also, there is no indication to what extent the final conclusions depend in this arbitrary choice. Further, the definition of 'uncertainty' in extremes is only explained in the caption of Figure 2, and I wonder why such great complexity is needed after all. Why not simply analysing the very highest/lowest monthly precipitation, runoff and ET sums at each grid cell, across models and forcing datasets?

Moreover, it remains unclear if absolute values or anomalies (i.e. with removed seasonal cycle) are used. In the case of absolute values, high ET extremes will necessarily occur in summer and while this is not always the case for extreme precipitation, this would lead to a (unwanted) de-coupling of the variables with this analysis design.

Concerning the low extremes, I am not sure how much sense this makes for precipitation. Lets say in a dry grid cell precipitation is zero in most of the analyzed months, does it make sense then to determine such months as low precipitation extremes?

(2) I do not agree with referring to MSWEP as a 'gold standard', and with statements like 'the best global evapotranspiration products (Martens et al. 2017)' or 'simulation results from the earth2observe project [...] driven by the best available published precipitation observations'. While these products are certainly state-of-the-art, I doubt that they will be 'the best' (based on what measure?) in all regions and at all times. As for the reference precipitation used in this study, it could be a more fair alternative to use the ensemble mean across the considered precipitation products.

(3) I think the linearity assumption made in Figures 4-6 is not justified, such that the linear regressions are no suitable way to analyze these point clouds. Further, displaying the point cloud envelopes is misleading, as these envelopes is likely dominated by outliers/extremes, and do not necessarily reflect actual relationships. Instead, why not use a 2D density plot here, and climate-regime-based moving average lines to summarize the results?
* * *
Specific comments:

- section 3.1, line 13, and caption of Figure 3, and elsewhere: the authors sometimes refer to 'increases' while also decreases are found in some regions

- epsilon is used twice, in section 2.1, line 21, and then in section 2.2, line 10

- section 2.1, line 22: '20 mm annual precipitation' - does this refer to multi-year means, or to individual years

- section 2.1, line 23: abbreviation SD not defined

- section 2.1, line 25: replace 'runs' with 'simulations'

- section 2.2, line 2: 'simulator' is not defined

- section 2.2, line 18: I think here you mix up i with j (?)

- results section, and figure captions: instead of using X as subscript and then referring to ET or runoff, you could replace the X with Q or ET

- Figure 2: numbers on color bar are very small

- Figure 3, caption: you mention a 'run' here, but these are just precipitation products and no model simulations

- Figures 4-7: legends missing

[Figure]

---

## Author Comment (AC3) · 3 Apr 2019

Review of Marthews et al. "A global scale evaluation of extreme events in the earth2observe project" The authors use model simulations from the earth2observe project to study the sources of uncertainty in simulated runoff and evapotranspiration (ET). Model simulations from this project are well chosen for this purpose as they are performed with (i) different precipitation forcing datasets and (ii) different land surface and hydrological models. Analysing these simulations, the authors compared the relative importance of the pre- cipitation forcing uncertainty with that of the model uncertainty for resulting runoff and ET extremes.
* * *
Recommendation: I think the paper should be rejected.

While the research question is interesting and relevant, and the model simulations are well suited for the purpose of this study, the applied methodology is too complex and hard to understand such that I am not sure about the robustness of the resulting conclusions.

-- RESPONSE -- Although this is a disappointing recommendation, thank you very much for the review and we hope very much that you will consider the responses below and, hopefully, we can convince you of the merit of this paper and our results.
* * *
General comments:

(1)  As mentioned above I do not understand (the purpose of) the methodology applied in this study, even after carefully reading it many times.

While the focus on extremes is not explained or motivated, I also do not see why/how 10% of a 14-year time series can already be considered extreme. Also, there is no indication to what extent the final conclusions depend in this arbitrary choice. Further, the definition of 'uncertainty' in extremes is only explained in the caption of Figure 2, and I wonder why such great complexity is needed after all. Why not simply analysing the very highest/lowest monthly precipitation, runoff and ET sums at each grid cell, across models and forcing datasets?

-- RESPONSE -- Firstly, we apologise that we hadn't fully motivated the approach that we took for this analysis. We have amended the first paragraph of the introduction to include reference to the IPCC Special Report on extreme events and have highlighted the importance of looking at these events.

        Secondly, extreme events exist on a continuum so some kind of definition is always required in a study like this (heavy rainfall in the UK would be considered normal in the Philippines, etc.). It is very standard to choose 10% as a threshold (a Q10/Q90 method) for extreme events (e.g. "The Intergovernmental Panel on Climate Change (IPCC) suggests that "rare" means in the bottom 10% or top 10% of severity for a given event type in a given location" on https://www.encyclopedia.com/environment/energy-government-and-defense-magazines/extreme-weather ) so we have added a reference to IPCC (2004) to Section 2.1 where we specify this (it was not a focus of this paper to try to quantify the uncertainty related to the choice of 10% here). The text clarifying our definition of uncertainty has been taken out of the legend to Fig. 2 and added as a sentence to Section 2.1 as well.

        Finally, "simply analysing the very highest/lowest monthly precipitation ..." is unfortunately simply not appropriate in an analysis at the global level: precipitation distributions do not only change in terms of mean and variance from place to place, but also change in terms of the shape of the distribution, i.e. skewness and bimodality. In order to carry out an analysis that covers all biomes from rainforest to desert, as we have done here, we need to use statistical methods, and the techniques we

have used are no more complex than used in comparable studies: in fact, although the use of ensemble methods brings in some complexity, the actual basic stats involved is nothing more complicated than a standard deviation of occurrence data.

5   Moreover, it remains unclear if absolute values or anomalies (i.e. with removed sea- sonal cycle) are used. In the case of absolute values, high ET extremes will necessarily occur in summer and while this is not always the case for extreme precipitation, this would lead to a (unwanted) de-coupling of the variables with this analysis design.

10  -- RESPONSE -- We used neither absolute values nor anomalies: we have been clear throughout the paper that our analysis was based on *occurrence data*: in any particular gridcell we get the distribution of e.g. precipitation from MSWEP (which gives us a baseline) and then instead of considering an absolute value (e.g. 50 mm rainfall) or anomaly (e.g. 50 mm minus the mean for that gridcell), we compare to the normal distribution and note whether an extreme event has occurred (1 or 0). It is then

15  the occurrence numbers that are analysed/averaged. We believe this is the best way to analyse data that comes from widely disparate biomes with differing distributions of precipitation, ET or runoff. The analysis was also deliberately carried out month by month (e.g. comparing to a baseline calculated from all the Februaries in the 14 year MSWEP dataset) in order to exclude any spurious matching of e.g. winter months to summer months, which accounts perfectly for the de-coupling mentioned here.

20

Concerning the low extremes, I am not sure how much sense this makes for precipi- tation. Lets say in a dry grid cell precipitation is zero in most of the analyzed months, does it make sense then to determine such months as low precipitation extremes?

25  -- RESPONSE -- Please note in Section 2.1 we state that we masked out all gridcells with extremely low rainfall exactly to exclude this possibility.

I do not agree with referring to MSWEP as a 'gold standard', and with statements like 'the best global

evapotranspiration products (Martens et al. 2017)' or 'simulation results from the earth2observe project [...] driven by the best available published precipitation observations'. While these products are certainly state-of-the-art, I doubt that they will be 'the best' (based on what measure?) in all regions and at all times. As for the reference precipitation used in this study, it could be a more fair alternative to use the ensemble mean across the considered precipitation products.

-- RESPONSE -- We stand by these statements: we use "best" to mean "best available product/model at the current time" (which we do not feel implies "best in all regions at all times"). We do not believe that it would be better to take a simple mean across all the considered precipitation products: they are not all comparable because they have varying levels of processing, as we have highlighted in Table 2. The high level of sophistication in the MSWEP reanalysis and the thought and consideration they have put into producing that product and correcting for observational problems that occur in other products we believe quite legitimately support our phrase "gold standard" (although please note this is our phrase and does not come from the MSWEP team).

(1) I think the linearity assumption made in Figures 4-6 is not justified, such that the linear regressions are no suitable way to analyze these point clouds. Further, displaying the point cloud envelopes is misleading, as these envelopes is likely dominated by outliers/extremes, and do not necessarily reflect actual relationships. Instead, why not use a 2D density plot here, and climate-regime-based moving average lines to summarize the results?

-- RESPONSE -- We initially did use 2D density plots here, but the extremely large number of points (and substantial overlap) served to obscure the message that we were trying to communicate with these figures. Although we do accept that displaying the envelopes draw attention away from mean values towards the extremes, we feel that in a paper focused on extreme event analysis that this is not an inappropriate approach to take.

We do accept that applying a linear fit to these data is simplistic, and a number of alternatives were experimented with during the course of the analysis we carried out in this paper. However, applying more

sophisticated methods did not seem to be legitimate given that the only conclusions we were drawing from these figures was whether or not the trend was an increase or a decrease moving from left to right. We certainly do not contend at any point that the distribution of points is linear in theory: we just included these lines to indicate the trend, which is not clear to the eye from the envelopes (because they don't show 5 the point cloud) or from the point clouds themselves (because they overlap too much and would have had to have been separated into individual plots, which for space reasons we didn't want to do)
* * *
Specific comments:

10  - section 3.1, line 13, and caption of Figure 3, and elsewhere: the authors sometimes refer to 'increases' while also decreases are found in some regions -- RESPONSE -- We have checked these statements and they are correct: please note that when we say alpha "increased with precipitation", this means it correlates positively with precipitation, which is unrelated to areas of blue versus green on the associated maps in the same figure.

15  - epsilon is used twice, in section 2.1, line 21, and then in section 2.2, line 10 -- RESPONSE -- Thank you for spotting this! Corrected.

- section 2.1, line 22: '20 mm annual precipitation' - does this refer to multi-year means, or to individual years -- RESPONSE -- This is indeed the MSWEP multi-year mean (we have now added this information in parentheses - thanks)

20  - section 2.1, line 23: abbreviation SD not defined -- RESPONSE -- "standard deviation" has been added in

- section 2.1, line 25: replace 'runs' with 'simulations' -- RESPONSE -- Thank you for spotting this! Corrected (and one occurrence of "runs" in the discussion too)

- section 2.2, line 2: 'simulator' is not defined -- RESPONSE -- "simulator" replaced with "simulator 25 model"

- section 2.2, line 18: I think here you mix up i with j (?) -- RESPONSE -- Thank you for spotting this! Corrected.

- results section, and figure captions: instead of using X as subscript and then referring to ET or runoff,

you could replace the X with Q or ET -- RESPONSE -- In an earlier draft we did try this, but the large number of "Q or ET"s that necessarily have to occur in the text we felt obscured the message we were trying to write.

- Figure 2: numbers on color bar are very small -- RESPONSE -- Colour bar size increased by 10%

5 - Figure 3, caption: you mention a 'run' here, but these are just precipitation products and no model simulations -- RESPONSE -- Thank you for spotting this! Corrected.

- Figures 4-7: legends missing -- RESPONSE -- We do state in the legends "Points on the scatter plots are coloured according to latitudinal zones (Fig. 1)", which we hope is sufficient and saves having Fig. 1 as an inset on each of these figures.

10

---

## Author Response (AR2)

**"A global scale evaluation of extreme events in the *eartH2Observe* project" *by* Toby R. Marthews et al.**

**Editor Decision: Publish subject to revisions (further review by editor and referees)** (07 Oct 2019)
by Patricia Saco

Comments to the Author:

We have now received useful comments from two referees. Based on my own careful assessment of the revised paper and the response letter, I agree with the reviewers that there are still some aspects that need to be addressed before this manuscript can be considered for publication. Though the authors have provided detailed responses to the referee comments, the manuscript will be further improved by integrating some of the material of these responses into a revised submission.

Thank you very much for this feedback and we hope very much that our resubmission addresses all outstanding concerns in full.

In particular, the revised manuscript could be improved by addressing some of the concerns of the referees:

1) Please include a more clear justification of the methodology emphasizing its appropriateness and its advantages over using a simpler methodology (this will address one of the concerns of reviewer #2, but please note that lack of clarity was also pointed out by the comments reviewer #1 on the original submission).

We have given detail in our responses to the reviewers, but we believe that we have much improved the presentation and justification of the methodological approach with our additional text, our new Figure (Fig. 2) and the worked example that is included in that figure.

2) Please analyze/discuss the robustness of results as suggested by reviewer #2, or alternative possible limitations of the study.

Please see added text and specific responses to all the individual reviewer comments below (Reviewer #3 first, followed by Reviewer #2 further below - all given as track changes on the previous, interactive review on *HESSD*).

3) Regarding the linearity assumption made in Figures 4-6. It would be good to add a very short discussion (a couple of sentences) similar to that included in the authors response, to clearly state

that the intent is not to suggest that the points in the figures follow a linear trend.

Thank you for this and we have followed this advice, adding in a sentence to each figure legend (based on the form of words given by the reviewer) stating clearly that we did not intend to imply that the underlying processes here are linear.

Though these are some of the main concerns that need to be addressed, please note that it is important to address all the reviewers' comments as this will help improve the contribution.

Thank you very much for the opportunity to make a second response to these reviewer comments. We have revised all sections of the paper as well as the reviewer comments (and we have even rechecked our responses to the first original reviewer as well). We hope very much that with these changes this article might still be considered for publication in *HESS*.

Very many thanks for your patience with this article and for your helpful feedback at all points.

Best regards,

Toby Marthews, Eleanor Blyth, Alberto Martínez and Ted Veldkamp.

30th October 2019.

**Response to interactive comment on* "A global scale evaluation of extreme events in the *eartH2Observe* project" *by* Toby R. Marthews et al.**

**Anonymous Referee #3 [= Anonymous Referee #2 from the first round of review]**
Received 7 October 2019
Second review of Hydrol. Earth Syst. Sci. Discuss., https://doi.org/10.5194/hess-2018-622, 2019.

The authors have implemented some of my suggestions in the revision of the manuscript. However, my main concerns with the complex methodology and my corresponding doubts regarding the robustness of the results and conclusions have not been addressed satisfactorily.

Therefore, I still recommend rejection of the paper.

Firstly, an apology to this reviewer: we can see that this response was submitted to *HESS* on 27th June, but we only received it on 7th October after the second reviewer response had also been received. We have responded as quickly as possible to the concerns raised and we hope that the delay will not count against us in this case.

We thank this reviewer for having given our manuscript due consideration both for the first review and also in this re-review, and we appreciate greatly that he/she has put in a considerable amount of time in formulating these comments with the motivation of helping us to improve this manuscript. In the light of these further comments, we have made many changes to the text to accommodate the suggestions raised and we hope very much that the quality of the paper has now been raised up to what is expected from a *HESS* article.
* * *
General comments:

This is not an easy decision to reach for me as I still appreciate the interesting research question, and the unique and very suitable dataset which the authors investigate. However,

(1) As mentioned above, I feel that none of the 3 major comments from my first review have been satisfactorily addressed in the revision of the manuscript. And this is even though I recommended rejection, which should already indicate that these are serious shortcomings in my opinion.

(2) I realize that some of my criticism in comment (1), namely the latter points on absolute values versus anomalies, and on low precipitation extremes, was maybe not fully justified as this was explained in the manuscript.
Thanks to the authors for pointing me to the corresponding paragraphs in the responses. Nevertheless, such misunderstandings could have been used as a motivation to try to further clarify these issues in the manuscript. Further, my main concerns in comment (1),

the complex methodology and the definition of extreme events, has not been addressed in the revision process. The authors could not convince me of the necessity of the cumbersome methodology.

Also, Comments (2) and (3) have not been much addressed overall.

(3) Even if the authors, for different reasons, disagree with most of my main suggestions, they could have done some sensitivity testing to show that the conclusions are robust with respect to my concerns. For this purpose, they could have added some results obtained with alternative methodologies (or reference precipitation).

The major issues referred to in the previous review are:

(1) *Our use of a 10% threshold to define extreme events and the suggestion that our method was overall more complicated than justified by the data and objectives of our analysis*
    We justified before our use of the 10% threshold (the accepted standard of the IPCC) and for method clarification we have inserted a whole new figure, supporting text and a worked example to justify our analysis approach (the new Fig. 2). We very much hope that this will be enough to convince this reviewer that our approach was indeed justified by our data and the objectives of our research as stated.

(2) *Referring to MSWEP as a 'gold standard' and to other comments about 'the best' evapotranspiration products, etc.*
    We have now removed all these statements about 'gold standard' and 'best'.

(3) *The linearity assumption made in our Figures 4-6 [now Figs. 5-7]*
    We would like to emphasise that by applying regression lines to these plots we only intended to support the statement that there was a general trend in our data from left to right. The regression fit applied was not used in any subsequent analysis and was only intended for visualisation of the trend. One of the plots referred to by this reviewer comment is included below for clarity: this reviewer suggested to "use a 2D density plot here, and climate-regime-based moving average lines [instead of regression lines]", however the other reviewer suggested instead to leave the combination plot as it is and simply to "add a very short discussion (a couple of sentences) similar to that included in the authors' response, to clearly state that the intent is not to suggest that the figures suggest that the points follow a linear trend"
    Having tried various options here, we would like to follow the second reviewer's suggestion for the following reasons: (i) firstly, space - each of Figs. 4,5,6 are currently composite 8-plot figures (with the first 4 plots of each being a scatter plot like the one below) so to split each scatter plot into 5 density plots (for each of the latitudinal zones of Fig. 1) would make each of Figs. 4,5,6 into a composite 24-plot figure and we believe that this would put us substantially beyond the length restrictions of *HESS* articles. Secondly, (ii) to replace the regression lines with moving-average lines, the righthand halves of these lines all disappear into the y=0 line and become indistinguishable and it is no longer possible to see the basic message communicated by the regression lines that there is a (statistically significant) general trend to the right. We would very much like to show more of our data in these plots by plotting the complete point clouds, but we believe that it is simply not

possible given the space constraints of this article, therefore we have opted instead to add in the sentence to each figure legend "Linear regression lines for each latitudinal zone indicate the trend as precipitation increases within each zone (all regressions were significant at the 1% level), although n.b. we do not contend in any way that the distribution of points shown is linear: these lines simply indicate a trend that is not clear to the eye from the envelopes displayed (which do not show the complete point cloud)" as requested by the second reviewer.

[Figure]

Instead of such actual changes/additions to the manuscript and/or the supplementary material, the authors are in many cases providing explanatory justification in their responses to my review comments.
And even in case of some comments which the authors decided not to consider as requested, they could still have included (some of) that justification into the manuscript such that all readers become aware of their arguments.

We like to believe that we have indeed now included the extra justification referred to here.

On a final point, we would like very strongly to thank this reviewer for raising all these issues. Including the smaller specific points at the end of the original review, we have benefited from a large number of well-chosen comments here and the manuscript has been changed throughout as a result of these insightful responses - many of which have brought up important aspects of the method and results that we failed to explain clearly in our original submission. Our manuscript is very much improved as a result of the raising of these concerns and it is now clear that we were indeed very much too brief on several parts of our method explanation. We apologise again that our lack of explanation caused this to be a more difficult paper to review than it could have been.

Best regards,

Toby Marthews *et al.*

**Anonymous Referee #2 [reviewer RC2 on https://www.hydrol-earth-syst-sci-discuss.net/hess-2018-622/]**

Review of Marthews et al. "A global scale evaluation of extreme events in the earth2observe project"

The authors use model simulations from the earth2observe project to study the sources of uncertainty in simulated runoff and evapotranspiration (ET). Model simulations from this project are well chosen for this purpose as they are performed with (i) different precipitation forcing datasets and (ii) different land surface and hydrological models. Analysing these simulations, the authors compared the relative importance of the pre- cipitation forcing uncertainty with that of the model uncertainty for resulting runoff and ET extremes.

———————————

Recommendation: I think the paper should be rejected.

While the research question is interesting and relevant, and the model simulations are well suited for the purpose of this study, the applied methodology is too complex and hard to understand such that I am not sure about the robustness of the resulting conclusions.

-- RESPONSE -- Although this is a disappointing recommendation, thank you very much for the review and we hope very much that you will consider the responses below and, hopefully, we can convince you of the merit of this paper and our results.

———————————-

General comments:

(1) As mentioned above I do not understand (the purpose of) the methodology applied in

this study, even after carefully reading it many times.

While the focus on extremes is not explained or motivated, I also do not see why/how 10% of a 14-year time series can already be considered extreme. Also, there is no indication to what extent the final conclusions depend in this arbitrary choice. Further, the definition of 'uncertainty' in extremes is only explained in the caption of Figure 2 [now Fig. 3], and I wonder why such great complexity is needed after all. Why not simply analysing the very highest/lowest monthly precipitation, runoff and ET sums at each grid cell, across models and forcing datasets?

-- RESPONSE -- Firstly, we apologise that we hadn't fully motivated the approach that we took for this analysis. We have amended the first paragraph of the introduction to include reference to the IPCC Special Report on extreme events and have highlighted the importance of looking at these events.

Secondly, extreme events exist on a continuum so some kind of definition is always required in a study like this (heavy rainfall in the UK would be considered normal in the Philippines, etc.). It is very standard to choose 10% as a threshold (a Q10/Q90 method) for extreme events (e.g. "The Intergovernmental Panel on Climate Change (IPCC) suggests that "rare" means in the bottom 10% or top 10% of severity for a given event type in a given location" on https://www.encyclopedia.com/environment/energy-government-and-defense-magazines/extreme-weather ) so we have added a reference to IPCC (2014) to Section 2.1 where we specify this (it was not a focus of this paper to try to quantify the uncertainty related to the choice of 10% here). The text clarifying our definition of uncertainty has been taken out of the legend to Fig. 2 [now Fig. 3] and added as a sentence to Section 2.1 as well.

Finally, "simply analysing the very highest/lowest monthly precipitation ..." is unfortunately simply not appropriate in an analysis at the global level: precipitation distributions do not only change in terms of mean and variance from place to place, but also change in terms of the shape of the distribution, i.e. skewness and bimodality. In order to carry out an analysis that covers all biomes from rainforest to desert, as we have done here, we need to use statistical methods, and the techniques we have used are no more complex than used in comparable studies: in fact, although the use of ensemble methods brings in

some complexity, the actual basic stats involved is nothing more complicated than a standard deviation of occurrence data.

**Second response:** Making use of the suggestions given by the reviewer here (combined with similar comments on the same topic from the other reviewer), we have inserted a new figure Fig. 2 into the paper featuring a flowchart explanation of the quantities alpha, beta and epsilon, as well as a worked example of how these are calculated and combined in the paper. We have also added text to sections 2.1 and 2.2 that we hope explain much more clearly what we have done and why.

We hope now that the presentation of a worked example here makes it clear why we have had to consider quantities that, on the face of it, appear to be complicated: it is the specific format of the source data of our study that requires us to do so (assembled by the many collaborators of the eartH2Observe project), and the approach dividing clearly between data and model uncertainty recommended by our guiding textbook Oberkampf & Roy (2010).

When we submitted the paper we were very conscious of space limitations and therefore we only included the absolute minimum description of the method and analysis quantities used, however if we are allowed to have new Fig. 2 then we think it does outline the concepts of our uncertainty quantities and encapsulate their interrelationships in the briefest but unambiguous way possible.

Moreover, it remains unclear if absolute values or anomalies (i.e. with removed sea-sonal cycle) are used. In the case of absolute values, high ET extremes will necessarily occur in summer and while this is not always the case for extreme precipitation, this would lead to a (unwanted) de-coupling of the variables with this analysis design.

-- RESPONSE -- We used neither absolute values nor anomalies: we have been clear throughout the paper that our analysis was based on *occurrence data*: in any particular gridcell we get the distribution of e.g. precipitation from MSWEP (which gives us a baseline) and then instead of considering an absolute value (e.g. 50 mm rainfall) or anomaly (e.g. 50 mm minus the mean for that gridcell), we compare to the normal distribution and note whether an extreme event has occurred (1 or 0). It is then the occurrence numbers that are analysed/averaged. We believe this is the best way to analyse data that comes from widely

disparate biomes with differing distributions of precipitation, ET or runoff [this point has now been emphasised more clearly in new Fig. 2]. The analysis was also deliberately carried out month by month (e.g. comparing to a baseline calculated from all the Februaries in the 14 year MSWEP dataset) in order to exclude any spurious matching of e.g. winter months to summer months, which accounts perfectly for the de-coupling mentioned here [this last sentence has now been added to section 2.1 to emphasise this point].

Concerning the low extremes, I am not sure how much sense this makes for precipi- tation. Lets say in a dry grid cell precipitation is zero in most of the analyzed months, does it make sense then to determine such months as low precipitation extremes?

-- RESPONSE -- Please note in Section 2.1 we state that we masked out all gridcells with extremely low rainfall exactly to exclude this possibility.

(2)     I do not agree with referring to MSWEP as a 'gold standard', and with statements like 'the best global evapotranspiration products (Martens et al. 2017)' or 'simulation results from the earth2observe project [...] driven by the best available published precipitation observations'. While these products are certainly state-of-the-art, I doubt that they will be 'the best' (based on what measure?) in all regions and at all times. As for the reference precipitation used in this study, it could be a more fair alternative to use the ensemble mean across the considered precipitation products.

-- RESPONSE --
**Second response:** We have now removed the reference to MSWEP as a "gold standard" and all occurrences of the word "best" in this context.

(3)   I think the linearity assumption made in Figures 4-6 [now Figs. 5-7] is not justified, such that the linear regressions are no suitable way to analyze these point clouds. Further, displaying the point cloud envelopes is misleading, as these envelopes is likely dominated by outliers/extremes, and do not necessarily reflect actual relationships. Instead, why not use a 2D density plot here, and climate-regime-based moving average lines to summarize the results?

-- RESPONSE -- We initially did use 2D density plots here, but the extremely large number of points (and substantial overlap) served to obscure the message that we were trying to communicate with these figures. Although we do accept that displaying the envelopes draw attention away from mean values towards the extremes, we feel that in a paper focused on extreme event analysis that this is not an inappropriate approach to take.

We do accept that applying a linear fit to these data is simplistic, and a number of alternatives were experimented with during the course of the analysis we carried out in this paper. However, applying more sophisticated methods did not seem to be legitimate given that the only conclusions we were drawing from these figures was whether or not the trend was an increase or a decrease moving from left to right. We certainly do not contend at any point that the distribution of points is linear in theory: we just included these lines to indicate the trend, which is not clear to the eye from the envelopes (because they don't show the point cloud) or from the point clouds themselves (because they overlap too much and would have had to have been separated into individual plots, which for space reasons we didn't want to do)

——————————

Specific comments:

- section 3.1, line 13, and caption of Figure 3, and elsewhere: the authors sometimes refer to 'increases' while also decreases are found in some regions -- RESPONSE -- We have checked these statements and they are correct: please note that when we say alpha "increased with precipitation", this means it correlates positively with precipitation, which is unrelated to areas of blue versus green on the associated maps in the same figure.

- epsilon is used twice, in section 2.1, line 21, and then in section 2.2, line 10 -- RESPONSE -- Thank you for spotting this! Corrected.

- section 2.1, line 22: '20 mm annual precipitation' - does this refer to multi-year means, or to individual years -- RESPONSE -- This is indeed the MSWEP multi-year mean (we have now added this information in parentheses - thanks)

- section 2.1, line 23: abbreviation SD not defined -- RESPONSE -- "standard deviation" has been added in

- section 2.1, line 25: replace 'runs' with 'simulations' -- RESPONSE -- Thank you for spotting this! Corrected (and one occurrence of "runs" in the discussion too)

- section 2.2, line 2: 'simulator' is not defined -- RESPONSE -- "simulator" replaced with "simulator model"

- section 2.2, line 18: I think here you mix up i with j (?) -- RESPONSE -- Thank you for spotting

this! Corrected.

- results section, and figure captions: instead of using X as subscript and then referring to ET or runoff, you could replace the X with Q or ET -- RESPONSE -- In an earlier draft we did try this, but the large number of "Q or ET"s that necessarily have to occur in the text we felt obscured the message we were trying to write.

- Figure 2: numbers on color bar are very small -- RESPONSE -- Colour bar size increased by 10% - now increased by a further 50% in the Supp Mat

- Figure 3, caption: you mention a 'run' here, but these are just precipitation products and no model simulations -- RESPONSE -- Thank you for spotting this! Corrected.

- Figures 4-7: legends missing -- RESPONSE -- We do state in the legends "Points on the scatter plots are coloured according to latitudinal zones (Fig. 1)", which we hope is sufficient and saves having Fig. 1 as an inset on each of these figures.

**Response to interactive comment on* "A global scale evaluation of extreme events in the *eartH2Observe* project" *by* Toby R. Marthews et al.**

**Anonymous Referee #2**
Received 7 October 2019
Comment on Hydrol. Earth Syst. Sci. Discuss., https://doi.org/10.5194/hess-2018- 622, 2019.

================
I believe that the authors have addressed and/or provided a good rebuttal to the comments and concerns of the previous reviewers.

Thank you very much for this comment.

Though the authors have provided detailed responses to the referee comments, the manuscript would be further improved if some of these responses were added or discussed in the revised submission.

Thank you: we have indeed now added in a selection of the responses we made into the text itself where appropriate.

In particular, the revised manuscript could be improved by addressing some of the concerns of referees:
1. The selection of evaporation products. Please rephrase to clearly explain what you mean by best (i.e., not in all regions at all times), and I believe that "gold standard" is a bit of an overstatement and not needed in the context of the paper.

All references to "gold standard" and "best" have now been removed (see the other reviewer's General Concern #2).

2. The linearity assumption made in Figures 4-6. It would be good if the authors can add a very short discussion (a couple of sentences) similar to that included in the authors' response, to clearly state that the intent is not to suggest that the figures suggest that the points follow a linear trend.

Thank you for this comment and we refer to our response on the same point under the other reviewer's General Concern #3: we have followed exactly the advice given here.

Very many thanks for your time reviewing our manuscript: it is hugely appreciated.

Best regards,

Toby Marthews *et al.*

Although not specifically required at this stage in the review, we have revisited our responses to Referee #1 below in the light of the second round of review responses and we would like to modify some of our responses below:

**Anonymous Referee #1 [reviewer RC1 on https://www.hydrol-earth-syst-sci-discuss.net/hess-2018-622/]**

General comments The manuscript presents an analysis of a unique dataset that was produced from the eartH2Observe project. This dataset involves the simulation of sev- eral hydrologic variables from a number of state-of-art land surface/hydrologic models and using as forcing several satellite and reanalysis dataset. The scale of analysis is global and the focus is on the tails (i.e. low/high extremes) of evapotranspiration and surface runoff. Overall the work is very interesting and the dataset analyzed is very unique. Additionally, the fact that the analysis is performed at global scale provides im- portant information on the regional variability of findings. The manuscript is generally well written but there are certain parts (especially in the description of methodology) that require additional clarification and discussion. I provide some specific comments below that hopefully will help the authors to improve their manuscript.

-- RESPONSE -- Thank you very much for the review and we hope that the responses below are sufficient reply to these useful comments.

Specific comments

1. I believe that the title should be revised to better reflect the context of the paper. One of the main elements of you analysis is "uncertainty in identification of extreme events" but this is not reflected in the current title.
-- RESPONSE -- We agree and have added "uncertainty" to the title.

2. Abstract L17-19: I agree but given the focus of your analysis (i.e. identification of extremes) you should be more specific on what your results will allow to comment. For example, models can be quite robust in representing the main body of the distribution of hydrologic variables, which is actually very important for water resources applications. So I suggest to specifically refer again to representation of extremes.
-- RESPONSE -- We agree: we have now specified in the abstract that "Our results are important for highlighting the relative robustness of satellite products in the context of land

surface simulations *of extreme events*" as requested.

3. P3L3: For a multiregional evaluation of satellite precipitation over complex terrain, you may want to consider also Derin, Yagmur, et al. "Multiregional satellite precipitation products evaluation over complex terrain." Journal of Hydrometeorology 17.6 (2016): 1817-1836.
-- RESPONSE -- Thank you for this reference: it was indeed relevant and we have used it at two points in the revised text.

4. Information in Section 2.1 needs to further clarified. What do you mean by "base distribution"? Is this the reference for your EE/yr at each cell? Why you average the five runs and you don't consider each model independently? Do you repeat the same procedure for each product and then compare? Please clarify.
-- RESPONSE -- Thank you for pointing this out: we had not explained why a base distribution (or baseline distribution) is necessary. We have added to section 2.1 "Extremes for any particular variable may only be assessed in relation to an estimate of 'normal' conditions, and for this we took a baseline distribution of values ...". At this point we have not considered each model independently because we do that at a later point when comparing model uncertainty and data product uncertainty. Finally, yes we did exactly repeat the same procedure for each product (and each model) and compare: this is the basis of the definitions of alpha, epsilon and beta later on in the method.

5. It would be very useful to provide a graphical example to demonstrate the different uncertainty components that you describe in equations 1-3.
-- RESPONSE -- We have indeed included a graphical example of the different uncertainty components: this is Fig. 7 [now Fig. 8]. We have considered moving that figure to an earlier point in the paper, but it is difficult to do so because we need to explain more details of how it is calculated before presenting that figure.

**Second response:** Having revisited this response, we have come to the conclusion that it was not a fully adequate response and we apologise for this. The addition of Fig. 8 was good, we still feel, but it served only to clarify our results, not the description of our methodology. We have now also inserted a new Fig. 2 that presents the relationships between alpha, beta and epsilon in flowchart form and a worked example of how these quantities relate to each other. The definitions here are the product of a lot of thought based on methods outlined in Oberkampf & Roy (2010) and they are necessary for an uncertainty analysis carried out in a model ensemble context.

We apologise that something similar to Fig. 2 was not inserted into the paper before. Each step described there was actually implicit in the careful wording of our definitions of alpha, beta and epsilon before, however we do accept that we failed to draw attention to the full implications of these definitions for our analysis (e.g. the significance and interpretation of epsilon>1 and why a quantity like *DIU* cannot be used directly when considering more than one variable of differing units).

6. L24-28 are confusing. First, it is not clear why you consider $\varepsilon x,j > 1$ as an indicator of model amplification of uncertainty? Do you mean $\alpha x,j$ instead? Also if you want to identify the relative contribution of the different sources of uncertainty, why don't you take the ratio of $\alpha/\beta$ ?

-- RESPONSE --

We have added some text to section 2.2 ("In summary ... product only") that we hope clarifies this point and the relationship between alpha, epsilon and beta.

**Second response:** With the insertion of the new Fig. 2, we hope that this has become a lot clearer and it is more visually presented now.

In response to the suggestion that alpha could be an indicator of model amplification rather than epsilon, we have inserted into the text new sentence "This augmentation comes from two sources: firstly, a model ensemble can produce outputs with higher sensitivity to input precipitation e.g. through a significant nonlinear relationship between $X$ and precipitation in the majority of ensemble models ($\alpha$), but it must not be forgotten that higher uncertainty in the outputs may also come from the differences in non-precipitation dependencies inside these models, which may also be larger in magnitude than $DIU$ ($\beta$)", which we hope clarifies that alpha is only one aspect of model augmentation and in this analysis it is very important to consider both aspects: the influence of precipitation (which comes through in alpha), yes, but also the influence of non-precipitation factors (which generally come through in beta, as a dependence on 'choice of model'). Incidentally, this new sentence was itself borrowed (with thanks!) from the same reviewer's comment at #14 below where he/she brought up a similar point.

7. P6L6: "global average", why do you consider global average? It is not advisable since the average masks regional variability. Also "ET highs (58.1% vs 41.9%)", it is not clear what these numbers correspond to.

-- RESPONSE -- This is an introductory point at the start of section 3.1 which we then expand upon in more detail later. It is not irrelevant to point out that the alpha values are universally quite small and do seem to decline with increasing precipitation. the regional variability is displayed graphically in Figs. 4-6 [now Figs. 5-7].

8. P6L10 "$\alpha x,j < -1$", I believe you mean $\log(\alpha x,j)$.

-- RESPONSE -- The reviewer was correct (many apologies!): The text has now been amended to "$\alpha X,j < 0.1$, $\log_{10}(\alpha X,j) < -1$".

9. P6L19-23. Interesting findings, some additional comments are welcome here. For example, why "the magnitude of the increase reduced in wetter environments"?

-- RESPONSE -- We feel that it would be too speculative to include here any of the various theories that could explain why the magnitude of the increase is reduced in wetter environments. For example, there could be a saturation effect in the environment (but in the absence of soils or land use data we cannot be sure of this) or fast drainage could occur more often under more episodic rainfall (but we have no data on drainage patterns) or the

occurrence of convective cells might be very regionally specific (but these are not even visible on most remote sensing products). We have tried to be careful to stick to discussing points that are directly relevant to the results and data that we have presented and we hope in a later study to look at these trends in more detail, but we have omitted any discussion of this here.

10. P6L25: "The global mean value. . .is a measure of variability". How can a mean value tell you anything about variability? Please clarify/revise.
-- RESPONSE -- If the quantity in question (alpha) is itself a measure of variability, then the mean of alpha will still be a measure of variability even though we agree it will not contain any information about the variability of alpha itself. We have revised the wording here to avoid the apparent contradiction.

11. P6L25-30: In general, this part of the text is quite difficult to "digest". Please improve clarity.
-- RESPONSE -- We agree and thanks: we have removed the middle sentence, which we hope has improved the clarity of the paragraph.

12. P6L31: What do you mean by "internal model uncertainty"?
-- RESPONSE -- We have added in the explanation that this is "a measure of the diversity of the calculation methods used to derive $X$ between models".

13. P7L3-4: "…are more sensitive to precipitation extremes in wet environments". Be careful here, you should state "…more sensitive to precipitation uncertainty".
-- RESPONSE -- Corrected with thanks.

14. P7L15-16: I believe that there is a confusion here between model uncertainty and uncertainty propagation. This is a very important aspect and the authors should clarify it in their discussion. For example, even with zero model uncertainty, transformation of precipitation uncertainty to runoff uncertainty could potentially amplify as a result of the nonlinear transformation of rainfall-to-runoff.
-- RESPONSE -- We have specifically defined separate quantities for model uncertainty (beta) and uncertainty propagation (epsilon) and we believe that we have not confused the two issues in this paper: in fact, drawing attention to the difference is one of the overall points of the paper.

   If runoff is generally 1000-1500 mm/yr with 7 peaks/yr when precipitation inputs are 500-1000 mm/yr with 3 peaks/year, then output uncertainty differs not only in terms of absolute value (which can be a linear effect) but also in terms of distribution (a nonlinear effect). By focusing our study on extreme event occurrence, linear effects should be cancelled out (as long as the extremes are calculated in terms of an appropriate baseline for each quantity, which we have done), however of course there will be nonlinear effects that can give nontrivial values to epsilon (and alpha) even in the case of beta=0 because the number of peaks may still change. At no point in the paper have we assumed that this will not happen: in fact, we have accounted for this in all analyses.

At P7L15-16 we have simply stated that model uncertainty is usually greater than data uncertainty. We believe that the reviewer here does not like the implication that when model uncertainty is small then data uncertainty must be even smaller, and it was certainly not our intention to imply that. We have modified the text to say "when a set of models is under consideration, model uncertainty is usually greater than data uncertainty". To avoid the same implication we have added "in a simulation ensemble" to the start of section 4.2 as well.

15. The same point as in 14(above) should be considered in the discussion of section 4.2 (e.g. L26-27).
-- RESPONSE -- Please see last point #14.

16. P9L10: ". . .to improve prediction of water cycle quantities". Ok I agree but the analysis presented has not done anything on the quantitative aspect. Perhaps revise to "improve prediction of water cycle extremes"?
-- RESPONSE -- Thank you for the suggestion: changed to "extremes"

17. Section 4.3. (L15-22). The text here is relevant to work that is evaluating uncer- tainty and compares against observations. However, this is not the scope of your work. You isolate (correctly) the forcing and model uncertainty by considering as reference a model/forcing combination.
-- RESPONSE -- These comments are made in a section entitled "4.3 Sources of unquantified uncertainty" and we state clearly in the preceding sentence that we could not analyse these kind of situations in our particular study given the data available. However, we find these issues to be entirely within scope of this study and, in fact, we would have been remiss not to have mentioned them. We hope very much in a follow-up study to find some way to tackle these sorts of issues and we believe it is entirely appropriate to have a brief mention of them here.

18. Fig2 [now Fig. 3]: What is (a) and what is (b). Also, some of the explanation on the calculation of results could be added to text in manuscript as well.
-- RESPONSE -- The legend stated "a. Uncertainty in precipitation extreme highs and b. Uncertainty in precipitation extreme lows", which perhaps was not clear because we did not use parentheses on the (a) and (b), so parentheses have been added in. We agree that the explanatory text in the legend was perhaps too long and was mostly superfluous because the calculation is already described in the main text (section 2.1) so we have now omitted it.

19. Fig3 [now Fig. 4]. Similar comment on the explanation.
-- RESPONSE -- Unlike for Fig. 2 [now Fig. 3], the description of the calculation for Fig. 3 [now Fig. 4] is not repeated in the main text, but after reconsidering the legend we would like to argue that the amount of detail here is appropriate: the explanation here simply describes what the rows and columns of this multi-panel plot display and we do not see any way to abbreviate this without forcing the reader to hunt through the text for this description. Therefore we have left this text as it is and we hope the reviewer will either reonsider this comment or specify more precisely what change he/she would like us to make, please?.

20. Figure 4 [now Fig. 5]. I find this map very useful. It would be nice to provide for the other cases analyzed.
-- RESPONSE -- The other 3*3=9 maps from Figs. 4, 5 and 6 [now Figs. 5-7] were excluded before simply from space considerations. They have now been added.

21. Figure 7 [now Fig. 8]: "erros bars show SE". Do you mean standard error? And how the error is defined. Perhaps you refer to standard deviation instead?
-- RESPONSE -- We have left this text as it is: what was calculated here was standard error, which differs from standard deviation because you divide by the square root of sample size (the abbreviation SE is standard). The "averaged over 50°S to 50°N" earlier in the legend makes it clear that this is calculated across gridcells rather than time (i.e. sample size is the number of gridcells in this case).

[revised manuscript text omitted]

42    *Environmental Modelling & Software*, *23*(10), 4.